# Techno-economic assessment of effervescent tablet-based nanofluids

**Naser Ali** [1]*, **Husain Bahzad**[2], **Nawaf F. Aljuwayhel** [3], **Shikha A. Ebrahim**[3]

**1** Nanotechnology Applications Program, Energy and Building Research Center, Kuwait Institute for Scientific Research, Safat, Kuwait, **2** Department of Chemical Engineering Technology, Public Authority for Applied Education and Training, Safat, Kuwait, **3** Mechanical Engineering Department, College of Engineering and Petroleum, Kuwait University, Safat, Kuwait

* nmali@kisr.edu.kw

**Data availability statement:** All relevant data are within the manuscript and its Supporting information files.

**Funding:** This study was funded by the Kuwait Foundation for the Advancement of Sciences (KFAS) in the form of a grant [PN23-35EM-1837 to NA] and the Kuwait Institute for Scientific Research (KISR) in the form of a grant [EA112C to NA]. The funders did not play any role in the study design, data collection and

## Abstract

The techno-economics of producing effervescent tablet-based nanofluids in start-up industries was investigated. The thermophysical properties, equipment costs, consumables, and operational expenditures for the production methods were determined. Electrical costs for running the devices were assessed based on the lowest, average, and highest prices worldwide. Key financial metrics, including accumulated project cost, interest, annual cost, operating expenses, production cost, payback and discounted payback periods, and internal rate of return (IRR) were calculated for different interest rates and amounts of sold products. The thermal performance under laminar and turbulent flow conditions and cost effectiveness were determined. The results showed that the effervescent tablet-based nanofluid was more favourable than its conventional counterpart. This is because of the lower equipment cost for the effervescent tablet process compared to its counterpart. The interest rate showed significant influence on the annual and product costs for both types of projects. Electrical cost had a minimal impact on the previous costs for both project cases. In addition, the effervescent tablet-based project outperformed its counterpart in terms of payback period, discounted payback period, IRR, and thermal performance, making it an optimal choice for investors and decision-makers.

## 1. Introduction

Nanofluids are an advanced type of suspensions with superior thermophysical properties compared to their conventional counterparts [1]. They were initially proposed in 1993 by Masuda et al. [2], aiming to replace the commonly used working fluids in thermal energy cycles. Two years later, Choi and Eastman introduced the term 'nanofluid' to the scientific community [3] to specify that this category of suspensions is made from dispersed solid materials and that it does not exceed 100 nm in size. For many years, two routes have been adopted in the fabrication process [4]. The first route is via a one-step (or single-step) method, whereas the second route is via a two-step approach. In the one-step method, the nanoparticles are formed and dispersed in a single step. The advantages of this approach lie in the high level of physical stability of the dispersed particles and their low tendency to agglomerate. However, the disadvantage of this production method is that not all types of nanofluids can be

analysis, decision to publish, or preparation of the manuscript.

**Competing interests:** The authors have declared that no competing interests exist.

formed through this route. This implies that not all types of nanomaterials can be formed in a liquid to produce a specific nanofluid. Some well-known single-step approaches include laser ablation, microwave irradiation, vacuum evaporation onto a running oil substrate, physical vapour condensation, submerged arc nanoparticle synthesis, plasma discharge, and electrical explosion of wires [5]. Conversely, this two-step approach requires two stages to fabricate the nanofluid. The first stage entails obtaining dry nanoparticles, and, in the second stage, these particles are mechanically dispersed in a base fluid. This can involve the use of a magnetic stirrer, homogeniser, ultra-sonicator, milling device (ball or rod), or a sequence of previous equipment [6]. The disadvantages of the aforementioned approach are that any type of nanofluid can be formed, the equipment used in the fabrication process are inexpensive and easy to handle, and commercial nanoparticles are widely available in the market for users to purchase and use. Owing to these advantages, most researchers prefer the two-step approach when fabricating their nanofluids. This conclusion was also driven by the fact that the majority of available literature shows the use of the two-step approach in contrast to the one-step method [7,8]. Notably, nanofluids produced using the two-step approach are usually less stable than those formed using the one-step method [9,10]. However, this can be overcome by including surfactants or surface functionalisation of the nanoparticles before dispersion [11,12].

Currently, the scope of nanofluid research has expanded beyond fundamental studies to encompass a wide range of practical applications, including the successful development of medical diagnostic tracers [13], magnetic sealers [14], and lubricants [15,16]. Their use has also been explored in diesel oil [17,18], solar collectors [19], intercoolers [20], crude oil recovery [21], desalination systems [22], and air-conditioning systems [23,24]. The results of these investigations demonstrate the significant role of nanofluids in enhancing the efficiency of these applications.

However, despite their promising results, some of the challenges facing the commercialisation of such suspensions are as follows: high level of expertise required, need for sophisticated equipment and devices, and demand for electricity. Therefore, in 2023, Ali et al. [25] proposed a new method for producing nanofluids that depends on the effervescent tablet technology. In their fabrication approach, effervescent agents were mixed with the nanomaterial and then consolidated through a hydraulic compressor to form a tablet. Afterwards, the effervescent tablet is dropped into water, where it chemically reacts, causing the generation of $CO_2$ bubbles. The nanomaterials are then physically dispersed through the random motion caused by the generated $CO_2$ bubbles during their departure from the base fluid, after which the nanofluid is formed. Considering its inherent simplicity and lack of electrical requirements, the recently introduced production approach demonstrates superior convenience in the synthesis of nanofluids compared to traditional fabrication methods [26]. The proposed method represents an advancement, offering a more efficient and practical means of nanofluid production, and has proven to be successful for applications in hybrid vehicles [27] and coating [28]. However, an in-depth techno-economic assessment is required to compare conventional and effervescent tablet-based nanofluids before introducing these products to the market.

Inspired by the aforementioned information, this study investigates the techno-economics of producing effervescent tablet-based (MWCNTs) nanofluids and compares them to their conventional counterparts. The study is important for decision makers in start-up industries, as it demonstrates the thermal performance and cost-effectiveness of the two nanofluids production routes. To the best of the authors' knowledge, no such in-depth study has been conducted previously in the literature, which illustrates the novelty of this research. First, nanofluids of 0.15 vol. % of MWCNTs were produced using the two-step route and the effervescent tablet method. The thermophysical properties of the as-prepared suspensions were then determined. Next, the costs of equipment, consumables, and operational expenditures

were calculated. Electrical cost (EC) was calculated based on the lowest electrical cost (LEC), average electrical cost (AEC), and highest electrical cost (HEC) worldwide. This was done to provide insights into the electrical cost required to run nanofluid fabrication devices worldwide. In addition, the purchase of the capital and consumables was assumed in four scenarios: one-time payment, 10%, 20%, and 30% interest for 20 years. Thereafter, the payback period was calculated without and with a 10% discount rate. The accumulated project cost, accumulated interest, total annual cost, total operating expenses (OPEX), nanofluid production cost, internal rate of return (IRR), and payback period were determined. An uncertainty analysis of the investment cost and OPEX was then conducted. Finally, the thermal performances of the as-prepared nanofluids under laminar and turbulent flow conditions as well as their cost-effectiveness under these regimes were determined.

## 2. Research methodology

### 2.1. Nanofluid production and thermophysical properties

Two production routes were used to fabricate the nanofluids of 0.15 vol. %/L, precisely the conventional and effervescent tablet-based methods [25], as shown in Fig 1. In the conventional method, MWCNTs and sodium dodecyl sulfate (SDS), in a 1:1 weight ratio, were magnetically stirred for 5 min and then probe sonicated for 12 min to produce a suspension. The MWCNTs to SDS ratio was selected to aid in stabilising the dispersed particles, based on a previous study [29].

Conversely, in the effervescent tablet-based approach, a tablet is first formed, as shown in Fig 2, through hand mixing using a mortar and pestle, with the weight ratio of the MWCNTs, SDS, sodium carbonate ($Na_2CO_3$), and monosodium phosphate ($NaH_2PO_4$) being 1:1:6:20.4, respectively. It should be noted that the $Na_2CO_3$ and $NaH_2PO_4$ included are the effervescent

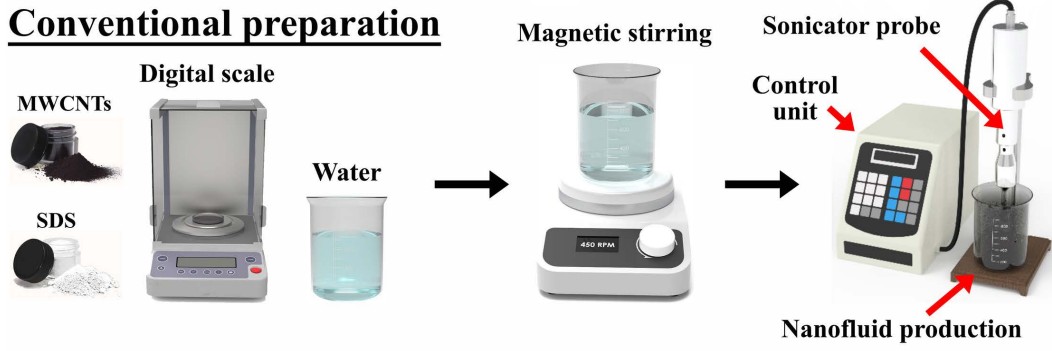

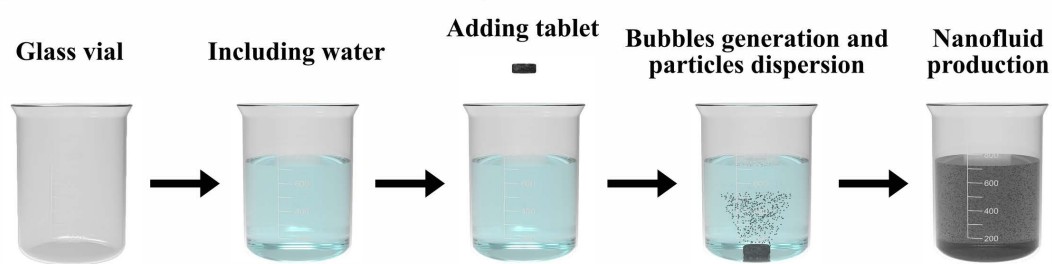

Fig 1. Illustration of the conventional two-step and effervescent tablet nanofluid production methods.

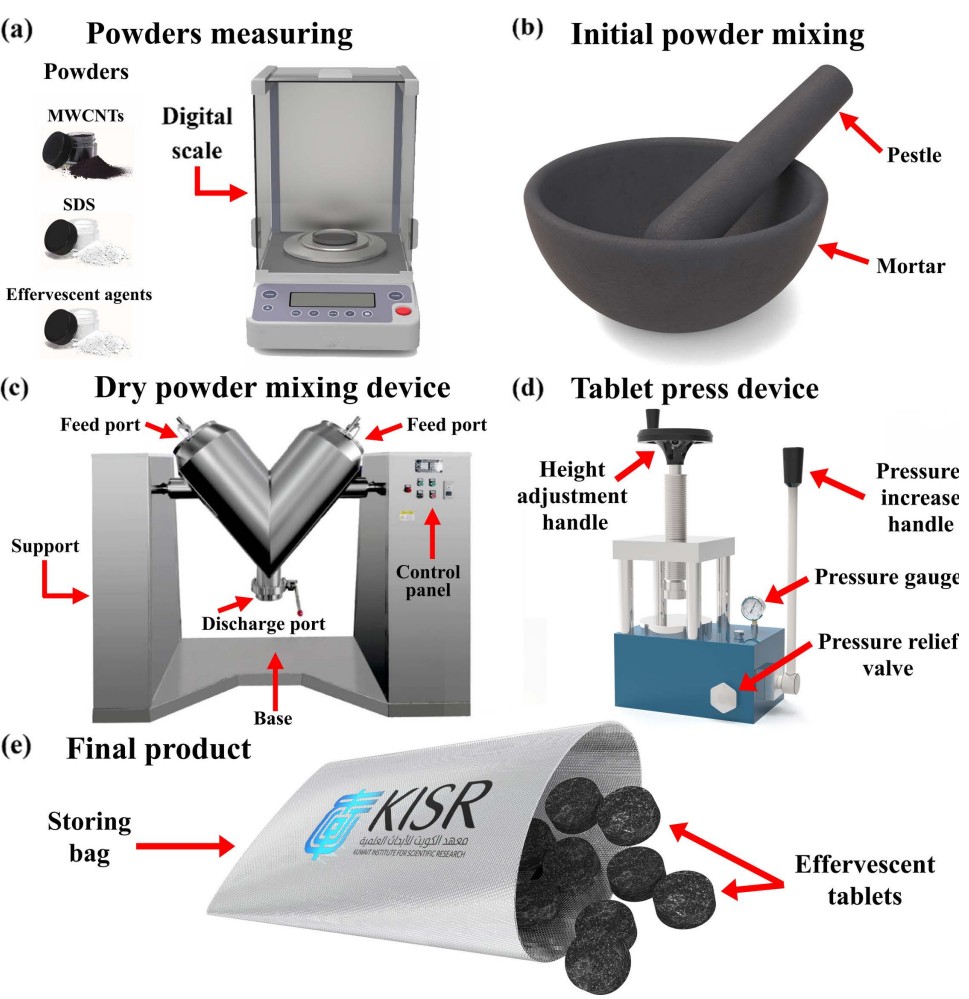

**Fig 2. Effervescent tablet fabrication for nanofluid production, where (a) shows powders measuring, (b-c) illustrate the mixing tools and device, (d) shows the consolidation of the as-mixed powders into tablets, and (e) presents the final product.**

agents that provide the effervescent effect in the tablets when added to water. The as-mixed powders were then intensively mixed for 20 min at 13 rpm using a dry powder mixing system of type XL-VH500. Subsequently, the as-mixed powders were consolidated at 100 kN into 25 mm tablets. The tablets were then added to the base fluid to initiate a chemical reaction, forming the nanofluid.

The thermophysical properties of both conventional and effervescent tablet-based nanofluid (NF) were determined through theoretical and experimental means from 20 °C to 40 °C, with a 5 °C increment. The nanofluid density and specific heat capacity were calculated according to the mixing theory correlations of Khanafer and Vafai [30] and Pak and Cho [31], as follows:

$$\rho_{NF} = \frac{m_{NF}}{V_{NF}} = \frac{\rho_{BF} V_{BF} + \sum_{j=1}^{x} m_{solid,j}}{V_{BF} + \sum_{j=1}^{x} \frac{m_{solid,j}}{\rho_{solid,j}}} \tag{1}$$

$$C_{NF} = \frac{m_{BF} C_{BF} + \sum_{j=1}^{x} C_{solid,j} . m_{solid,j}}{m_{BF} + \sum_{j=1}^{x} m_{solid,j}} \qquad (2)$$

where $\rho_{NF}$, $\rho_{BF}$, $m_{NF}$, $m_{BF}$, $V_{NF}$, $V_{BF}$, $C_{NF}$, and $C_{BF}$ are the nanofluid density, basefluid density, nanofluid mass, basefluid mass, nanofluid volume, basefluid volume, nanofluid specific heat capacity, basefluid specific heat capacity, respectively. The terms $\sum_{j=1}^{x} m_{solid,j}$, $\sum_{j=1}^{x} \frac{m_{solid,j}}{\rho_{solid,j}}$, and $\sum_{j=1}^{x} C_{solid,j} . m_{solid,j}$ in Eqs. 1 and 2 can be determined as follows:

$$\sum_{j=1}^{x} m_{solid,j} = m_{solid,1} + m_{solid,2} + ... + m_{solid,x} \qquad (3)$$

$$\sum_{j=1}^{x} \frac{m_{solid,j}}{\rho_{solid,j}} = \frac{m_{solid,1}}{\rho_{solid,1}} + \frac{m_{solid,2}}{\rho_{solid,2}} + ... + \frac{m_{solid,x}}{\rho_{solid,x}} \qquad (4)$$

$$\sum_{j=1}^{x} C_{solid,j} . m_{solid,j} = C_{solid,1} . m_{solid,1} + C_{solid,2} . m_{solid,2} + ... + C_{solid,x} . m_{solid,x} \qquad (5)$$

where $\rho_{solid,j}$, $m_{solid,j}$, $C_{solid,j}$, and $x$ are the solid materials density, solid materials mass, solid materials specific heat capacity, and the number of solid materials used (i.e., $x = 2$ (conventional), and 4 (effervescent tablet)), respectively. Table 1 shows the properties of the materials used in the previous calculations.

Conversely, the thermal conductivity ($\lambda$) and viscosity ($\mu$) of the basefluid along with the two types of nanofluids were measured using a transient hot wire (THW) of model L2 system (supplied by Thermtest Co.), and a rheometer of type HAAKE RheoStress 6000 (supplied by Thermo Scientific Co.), respectively. Three thermal conductivity readings were obtained for each sample, after which they were averaged. Thirty viscosity measurements were taken (10 s apart) at a shear rate of 200 s$^{-1}$ and then averaged.

## 2.2. Investment cost and OPEX

The total investment cost (TIC) was calculated according to the approach of 'Lang' [37], as follows:

$$TIC = TCIF \times EP \qquad (6)$$

where TCIF is the total capital investment factor listed in Table 2 and EP is the net cost of the devices and tools.

**Table 1. Density and specific heat capacity of starting materials.**

| Material type | Density (g/cm³) | | | | | Specific heat capacity (J/g.K) | | | | | Source |
|---|---|---|---|---|---|---|---|---|---|---|---|
| | 20 °C | 25 °C | 30 °C | 35 °C | 40 °C | 20 °C | 25 °C | 30 °C | 35 °C | 40 °C | |
| Water | 0.99822 | 0.99699 | 0.99562 | 0.99390 | 0.99217 | 4.1844 | 4.1816 | 4.1801 | 4.1799 | 4.1796 | [1,32] |
| MWCNT | 2.1 | 2.1 | 2.1 | 2.1 | 2.1 | 0.55 | 0.55 | 0.55 | 0.55 | 0.55 | Supplier, [33] |
| SDS | 1.01 | 1.01 | 1.01 | 1.01 | 1.01 | -0.725 | -0.725 | -0.725 | -0.725 | -0.725 | Supplier, [34] |
| Na$_2$CO$_3$ | 1.46 | 1.46 | 1.46 | 1.46 | 1.46 | 0.237 | 0.237 | 0.237 | 0.237 | 0.237 | Supplier, [35] |
| NaH$_2$PO$_4$ | 2.36 | 2.36 | 2.36 | 2.36 | 2.36 | 1.62 | 1.62 | 1.62 | 1.62 | 1.62 | Supplier, [36] |

**Table 2. Total investment cost based on Lang approach for the production process of the nanofluids.**

| Type of cost | Percentage of delivered instrument cost |
|---|---|
| Working capital (15% of the total capital investment). | 75 |
| Indirect cost (engineering supervision, construction, legal expenses, contractor's fee, contingency). | 126 |
| Direct cost (equipment installation, instrumentation, piping, electrical system, buildings, labours, and service facilities). | 302 |
| **Total capital investment factors** | **503** |

Furthermore, the list of devices and tools required to produce both conventional and effervescent tablet-based nanofluids is shown in S1 Table along with their costs. The prices of the equipment listed in S1 Table were obtained from the manufacturers.

The operating cost was calculated based on the cost of consumable materials and electricity consumed to run the devices, as represented in Eq. 7:

$$OPEX = C_c + E_c \tag{7}$$

where $C_c$ is the cost of the consumables listed in S2 Table, and $E_c$ is the electricity cost.

As such, three electricity price rates were considered: world's LEC ($0.007/kWh, Libya), AEC ($0.165/kWh), and HEC ($0.692/kWh, Solomon Islands) [38]. Furthermore, the consumables used in the analysis were assumed to correspond to the materials used in the fabrication of the MWCNT-based suspensions. S2 Table lists the consumable materials used as well as their costs, whereas S3 Table tabulates the amount of electricity required to run the devices used in the nanofluid production process.

## 2.3. Annual cost

The total annual cost ($TAC$) of a project was determined by considering the OPEX of the production process ($/year), and the annual value of the total investment cost of the production process ($A_c$) in $/year as follows:

$$TAC = A_c + OPEX \tag{8}$$

$A_c$ can be calculated according to the interest rate percentage ($i$) and lifetime of the project ($n$) in years, as follows:

$$A_c = (i+1)^n \left[ \frac{i}{(i+1)^n - 1} \right] \times TIC \tag{9}$$

Furthermore, $i$ was selected hypothetically to have the values of zero, 10%, 20%, and 30%. The $i$ equal zero value indicates that the settlement was made in a one-time payment, and thus, no interest was taken. Moreover, the project lifetime was assumed to be 20 years, which is commonly observed in chemical plants [39].

## 2.4. Nanofluid production cost and selling price

The production cost of each type of nanofluid (i.e., conventional and effervescent tablet-based), NFPC, was calculated based on the total annual cost of the project and the nanofluid annual production rate ($NF_P$) in L/year. This was performed using the following equations [40]:

$$NFPC = \frac{TAC}{NF_P} \tag{10}$$

It should be noted that $NF_P$ was summed to be 2 L on each working day for a full year. Thus, considering the total number of working days in the United States in 2023, this corresponds to 249 working days, after excluding public holidays and weekends. As such, $NF_P$ would be 498 L/year, or for simplicity, it can be assumed to be 500 L/year. Additionally, each nanofluid product was taken as 0.15 vol. %/L of MWCNTs. Furthermore, variations in the interest rate and electrical cost were considered when calculating the production cost for nanofluids. These products include those produced using the conventional two-step method and those fabricated using the effervescent tablet approach.

The selling price ( $SP$ ) of a manufactured product depends primarily on the break-even point [41]. Additional factors such as the established reputation of the manufacturing company, level of service offering, customer interest in the product, customer value, and existing competitors should also be considered. Most of the available literature [42–44] report that no single strategy exists for product pricing, especially for a newly developed product that has not been previously introduced to the market. As such, for simplicity, the selling price of the nanofluid products (i.e., both conventional and tablet-based) was calculated for each electrical price condition, based on the following:

$$SP = 3 \times NFPC_{avg} \tag{11}$$

where the $NFPC_{avg}$ is the average nanofluid production cost for all interest rate scenarios for both conventional and effervescent tablet-based nanofluids. Consequently, profit ($P$) can be determined as follows:

$$P = SP \times NF_P \tag{12}$$

## 2.5. Payback period and internal rate of return

The payback period ( $PP$ ), discount payback period ( $DPP$ ), and internal rate of return (IRR) were determined because of their crucial roles in planned future projects. This is because knowing the $PP$ and $DPP$ from the different available scenarios allows the investor to determine the duration required for the investment to generate sufficient cash flow to cover the initial cost and provide insights into the risk level. However, $DPP$ is a modified version of $PP$ that considers the time value of money by calculating the present value of future cash inflows. In general, shorter $PP$ and $DPP$ indicate a lower risk level because the planned investment recovers its cost faster, which means that fewer obstacles are likely to occur. Conversely, the IRR assesses how well the investment is likely to perform over time, thus providing a measure of the profitability of the project and guiding decision-makers in selecting the appropriate investment scenario. As such, the following equations were used to determine $PP$ and $DPP$ [45], assuming that 500 L (Case 1), 50 L (Case 2), and 10 L (Case 3) of nanofluids were sold per year:

$$PP = \frac{TIC}{(SP - NFPC) \times \text{LSN}} \tag{13}$$

$$DPP = \frac{\ln\left(1 - \frac{TIC \times r}{(SP - NFPC) \times \text{LSN}}\right)}{-\ln(1+r)} \tag{14}$$

where LSN and $r$ are the number of litres of nanofluid sold and the discount rate, respectively. For $DPP$ calculations, r is selected to be 10%. Conversely, the IRR was calculated with and without employing a 10% discount rate, using the same three previous LSN cases. The following equations were used for the calculation [46]:

$$IRR = \sum_{t=0}^{T} \frac{NCF_t}{(1+r)^t} - TIC = 0 \tag{15}$$

$$NCF_t = P - TAC \tag{16}$$

where T, t, and $NCF_t$ are the total number of time periods (in this case, 20 years), time period, and net cash inflow and outflow during a single period, t, respectively. A summary of the assumptions used in the economic analysis is shown in Table 3.

## 2.6. Investment cost and OPEX uncertainty analysis

In any potential investment, a degree of uncertainty is retained in the cost, mostly because of the estimation of the equipment cost [47]. For the novel process, the price of the equipment differs based on the retailers' offers. Thus, the effects of investment cost errors on nanofluid production costs at three fixed electricity prices (i.e., low, average, and high) were considered. Additionally, the IRR for one-time payment (0%), 10%, 20%, and 30% interest rate were calculated based on the investment costs. Therefore, the influences of the investment cost errors on the IRR at the three electricity prices were estimated. In addition, because of the daily variation in electricity prices, according to the energy supply and demand [48–50], the OPEX analysis carries some degree of uncertainty because of such variations in the electricity cost. Therefore, the impacts of OPEX errors on the nanofluid production cost were investigated. These errors affect the net cash flow, as shown in Eq. 13. Consequently, the effects of electricity cost errors on internal rate of return were studied. The sensitivity of the investment cost and electricity cost errors to the nanofluid production cost as well as the internal rate of return were determined by following an approach similar to that performed by Bahzad et al. [51]. The nanofluid production cost and the IRR after considering the error % in the TIC and electricity cost are denoted as '$NFPC_e$' and '$IRR_e$', respectively. Notably, these two parameters differ from those previously mentioned in Sections 2.4 and 2.5 (i.e., '$NFPC$' and 'IRR'),

Table 3. Summary of the assumptions used in the economic analysis.

| Economic parameter | Assumption |
|---|---|
| Interest rate | One-time payment (0%) |
| | 10% |
| | 20% |
| | 30% |
| Operation days in the year | 249 days |
| Electrical cost | $0.007/kWh (LEC) |
| | $0.165/kWh (AEC) |
| | $0.692/kWh (HEC) |
| Project lifetime | 20 years |
| Annual production rate | 10 L/year |
| | 50 L/year |
| | 500 L/year |

as they represent the original calculated values for the different cases. Furthermore, the TIC and the electricity cost was varied by ±20%, with a ±1% increment, for Case 1, which considered producing 500 L of nanofluid per year. For the TIC error %, the total annual cost was calculated using Eqs. 2 and 3 for the LEC, AEC, and HEC, respectively. The electricity cost was fixed at the values stated in Section 2.2. Consequently, the profits were estimated for the three electricity costs based on Eq. 12. Subsequently, the variation in the nanofluid production cost for each electricity cost was calculated using Eq. 10. In addition, the cash flows for the three electricity costs were determined using Eq. 16. Based on the determined cash flows, the IRR deviations were estimated for the LEC, AEC, and HEC using Eq. 15. For the electricity error % analysis, the TIC was assumed to be constant. For LEC, AEC, and HEC, the OPEX was estimated using Eq. 7. The total annual cost and error % in the nanofluid production cost were then determined using Eqs. 8 and 10. Moreover, the profits were estimated using Eq. 12, which was subsequently employed in Eq. 16 to obtain the cash flow. Consequently, the sensitivity of the $E_c$ in the IRR was estimated.

## 2.7. Thermo-economic analysis

From an economic perspective, the relationship between the heat-transfer fluid cost and its thermal performance is the cutting edge for considering its selection. As a working fluid can be utilised in a heat transfer application under laminar or turbulent flow conditions, both states must be considered. For a fluid to be beneficial under laminar flow condition, the ratio of viscosity enhancement coefficient to thermal conductivity enhancement coefficient ($C_\mu/C_\lambda$) should be > 4. Alternatively, a fluid would be beneficial under turbulent flow conditions if its Mouromtseff number (Mo) was found to be >1. As such, the $C_\mu/C_\lambda$ and Mo were calculated for both types of nanofluid, using Eqs. 17 and 18 [52,53]:

$$\frac{C_\mu}{C_\lambda} = \frac{\left(\mu_{NF} - \mu_{BF}\right)/\mu_{BF}}{\left(\lambda_{NF} - \lambda_{BF}\right)/\lambda_{BF}} \tag{17}$$

$$Mo = \left(\frac{\lambda_{NF}}{\lambda_{BF}}\right)^{0.67} \times \left(\frac{\rho_{NF}}{\rho_{BF}}\right)^{0.8} \times \left(\frac{C_{NF}}{C_{BF}}\right)^{0.33} \times \left(\frac{\mu_{NF}}{\mu_{BF}}\right)^{-0.47} \tag{18}$$

In addition, a high heat loss figure of merit ($FOM_{Heat\,losses}$) is preferred for heat transfer working fluids [54]. This figure of merit reflects the thermal transport capability of a fluid; hence, the higher it is, the better the thermal performance of the working fluid in heat-transfer applications, and vice versa. The previous value was calculated for the conventional and effervescent tablet-based nanofluids using Eq. 19:

$$FOM_{Heat\,losses} = \left(\frac{\lambda_{NF}}{\lambda_{BF}}\right)^{0.6} \times \left(\frac{\rho_{NF}}{\rho_{BF}}\right)^{0.34} \times \left(\frac{C_{NF}}{C_{BF}}\right)^{0.06} \times \left(\frac{\mu_{NF}}{\mu_{BF}}\right)^{-0.44} \tag{19}$$

Furthermore, including the cost of the working fluid as a factor (i.e., $/L), a thermos-economic analysis of both types of nanofluids was performed using a price-performance factor (PPF) metric under laminar ($PPF_C$) and turbulent ($PPF_{Mo}$) flow conditions using Eqs. 20 and 21 [55,56]:

$$PPF_C = \frac{C_\lambda/C_\mu}{Price\,(\$/L)} \tag{20}$$

$$PPF_{Mo} = \frac{Mo}{Price\ (\$/L)} \qquad (21)$$

## 3. Results and discussion

### 3.1. Suspension thermophysical properties

The influence of temperature on the density, specific heat capacity, thermal conductivity, and viscosity of the two types of nanofluids as well as their base fluids is shown in Fig 3. Both density (Fig 3a) and specific heat capacity (Fig 3b) decreased with increasing fluid temperature. However, the density of the effervescent tablet-based nanofluid was found to be higher than the conventional suspension and basefluid by 4% (at 20 °C) – 4.04% (at 40 °C) and 4.17% (at 20 °C) – 4.22% (at 40 °C), respectively. This increase corresponds to the additional material content (i.e., $NaH_2PO_4$ and $Na_2CO_3$) used in fabricating the effervescent tablet-based nanofluid compared to its conventional counterpart. Nevertheless, the change in temperature from 20 °C to 40 °C did not show a significant effect on the density as it only caused the property to reduce by 0.56%, 0.60%, and 0.61% for the effervescent tablet-based nanofluid, conventional suspension, and basefluid, respectively. Unlike the effervescent tablet-based suspension, which showed a reduction of 5.49% (at 20 °C) and 5.52% (at 40 °C) over its basefluid, the specific heat capacity of the conventional dispersion and its basefluid were seen to be very close to each other, with a variation of only 0.27% (at 20 °C and 40 °C). The thermal conductivity (Fig 3c) value over the designated temperature range (20 °C–40 °C) was found to be from 0.68181–0.7403 W/m. K, 0.62248–0.69391 W/m.K, and 0.5963–0.6294 W/m.K for the effervescent tablet-based nanofluid, conventional dispersion, and base fluid, respectively, with the tablet-based suspension being the highest. The effect of temperature change on the thermal property of the fluids was more pronounced. By contrast, the viscosity of the conventional nanofluid was higher than that of its counterpart by 26.33% (at 20 °C) and 10.08% (at 40 °C). The reason for this increase in the properties of the conventional suspension is likely because of the level of dispersed particles clustering and sedimentation formation, which interact with the number of collisions between the nanomaterial and water molecules. However, the increase in temperature enhances the Brownian motion of the dispersed nanomaterial and causes the interaction between the water molecules to decrease, resulting in a reduction in the viscosity of the fluid, as shown in Fig 3d.

### 3.2. Project cost and OPEX

Project cost analysis for both conventional two-step and effervescent tablet-based nanofluid production methods indicated that the effervescent tablet route is lower in cost by ~12.57% compared to its counterpart, as shown in Fig 4. This is mainly due to the variation in equipment cost between the two nanofluid production approaches, which is specifically 13.26% higher for the conventional route in comparison with the effervescent tablet process. The primary equipment causing such an increase in cost was the probe-type sonicator (i.e., the nanoparticles dispersing device), as denoted in S1 Table. Furthermore, the interest rate is significant in the project's accumulated cost, especially when considering that the interest rate has accumulated for 20 years. For example, the initial investment cost for the conventional method was $181,104.3, after which it increases to $425,448.85 (10% interest rate) and $1,092,373.6 (30% interest rate) at the end of the targeted duration (i.e., 20 years). Similarly, after adopting the effervescent tablet approach, the cost raised from $160,883.39 to $377,946.06 and $970,406.43 at the end of the 20 years period with 10% and 30% interest rates, respectively. In both aforementioned cases, the level of increase in the initial investment cost ranged from 134.92% (10% interest rate) to 503.17% (30% interest rate). Furthermore,

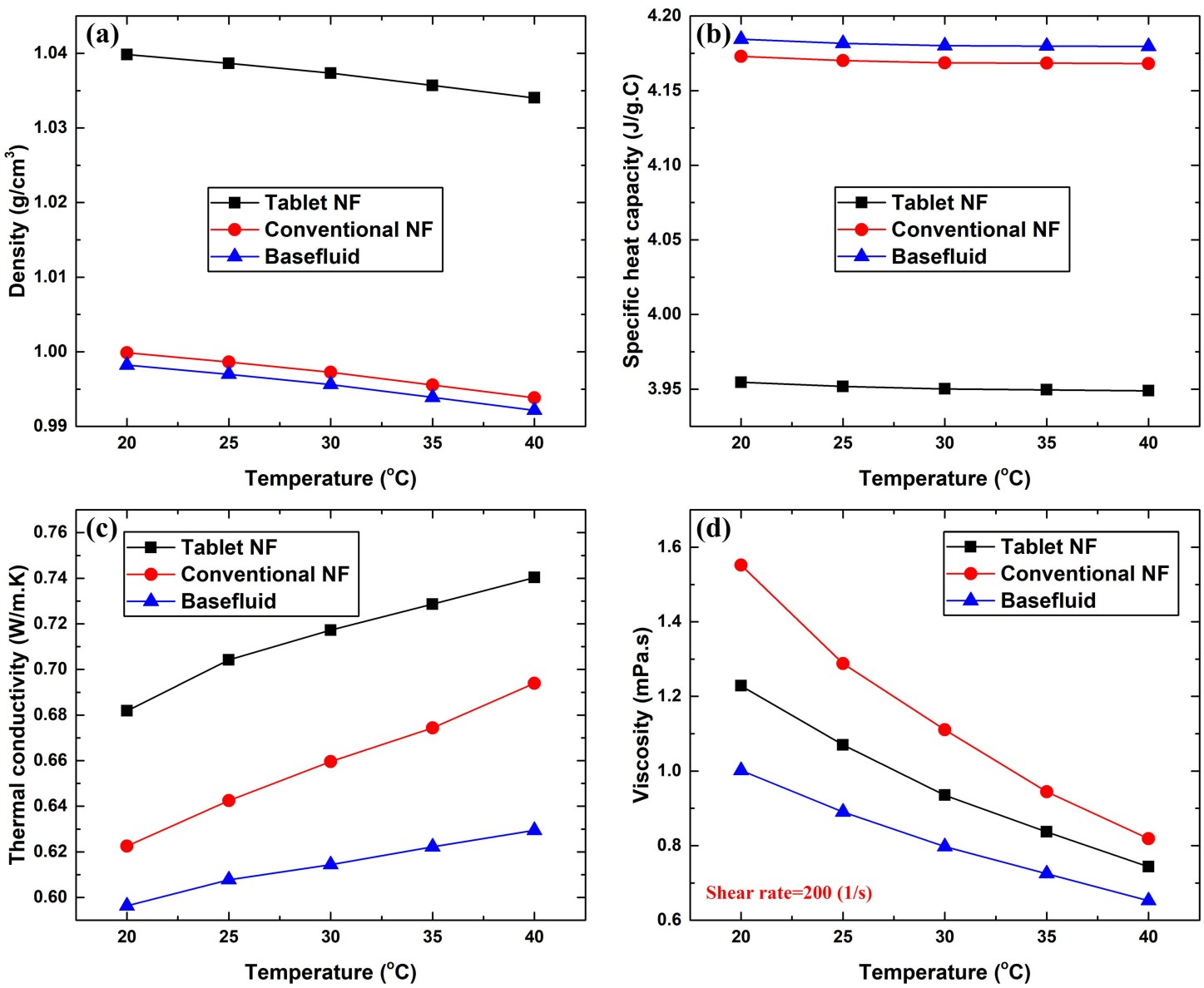

**Fig 3. Thermophysical properties of conventional and effervescent tablet-based nanofluid, where (a–d) illustrate the value of the density, specific heat capacity, thermal conductivity, and viscosity, respectively, from 20 °C to 40 °C.**

if decision makers considered settling their investment costs before the end date, they would make different savings based on the interest rate applied to their nanofluid production project. For instance, if the investment cost settling was made in years 15, 10, or 5, the savings would be 16.05%, 30.72%, and 43.85% for the 10% interest rate scenario, respectively. As for the 30% interest rate case, the savings would be 23.91%, 46.37%, and 65.97% when the settling is made in the 15th, 10th, and 5th years, respectively. Therefore, it is essential to monitor and gain insight into the accumulated interest values (Fig 5) so that decision-makers (or investors) can adjust their settlements according to their goals. In Fig 5, it should be noted that the values shown at time zero indicate that no interest rate was employed, and thus, this is considered as a one-time payment scenario. Further details regarding Figs 4 and 5 are provided in S4 Table.

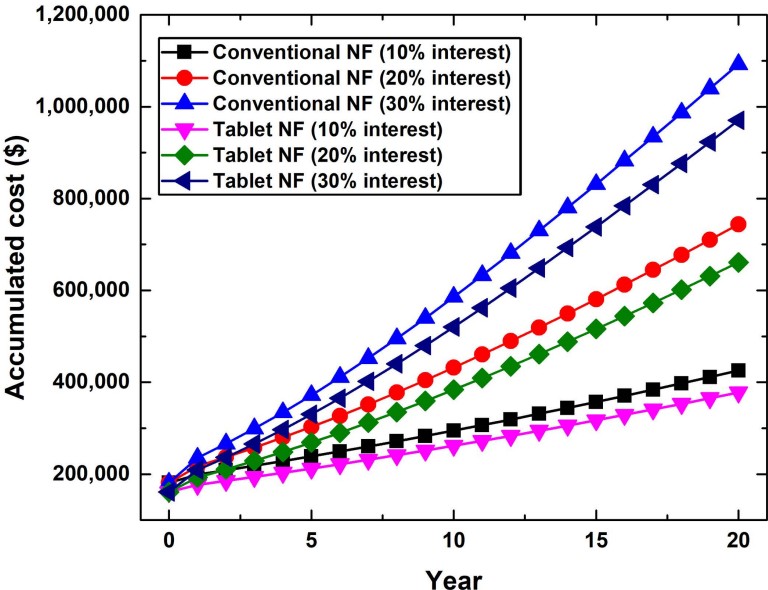

**Fig 4. Accumulated cost for the conventional two-step and effervescent tablet-based nanofluid production project at different interest rates.**

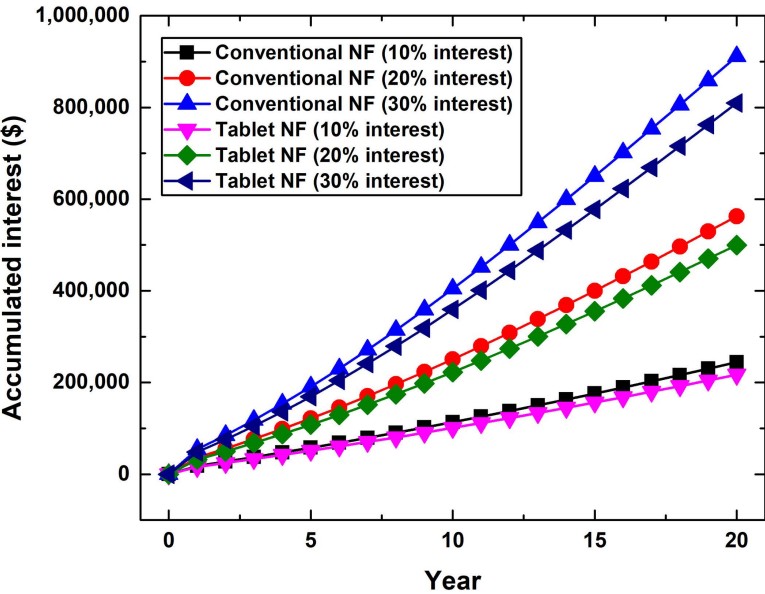

**Fig 5. Accumulated interest cost for the conventional two-step and effervescent tablet-based nanofluid production project at different set of interest rates.**

Conversely, the OPEX required for the effervescent tablet project was found to be significantly lower than that of the conventional nanofluid investment, precisely between 116.73% (HEC) and 121.39% (LEC), as illustrated in Fig 6. Moreover, the electrical cost effect was found to have a greater impact on the OPEX of the effervescent tablet project compared to that of the conventional two-step nanofluid investment. As an example, the OPEX in

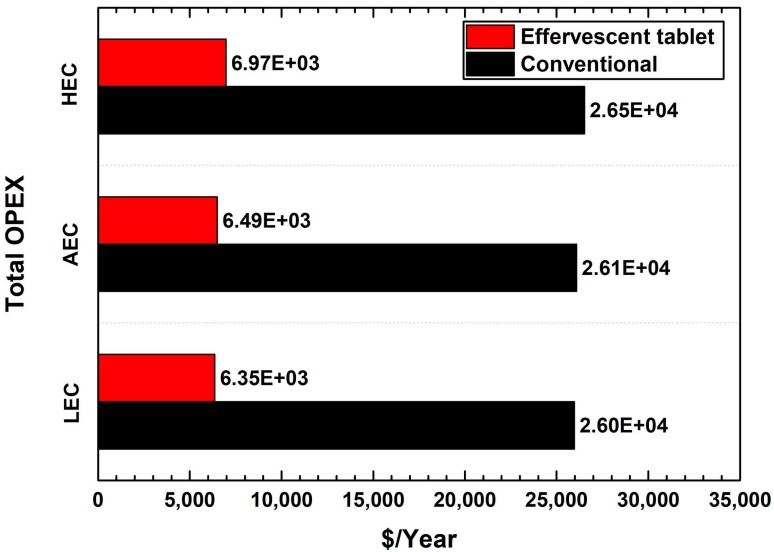

**Fig 6. Total OPEX cost per year for the different nanofluid production projects based on the electrical cost.**

countries with LEC and AEC will cost less by 9.31% and 7.08%, respectively, than those utilising HEC for the effervescent tablet-based nanofluid project. Alternatively, the electrical cost reduction in the conventional nanofluid project would only be approximately 2.09% and 1.61% for those having to pay an LEC and AEC, respectively, instead of the HEC. This finding demonstrates how different geographical locations can influence investors' decisions in selecting nanofluid production projects. Notably, the electrical cost represents 0.02% (LEC) to 2.09% (HEC) and 0.10% (LEC) to 8.98% (HEC) of the total yearly OPEX cost for the conventional and effervescent tablet production methods, respectively. Further information on the values used to generate Fig 6 is presented in S5 Table.

## 3.3. Total annual cost

The *TAC* was calculated for the two types of nanofluid production projects, as shown in Fig 7, and the detailed values are available in S6 Table. The one-time payment and interest rate scenarios are shown to favour the effervescent tablet approach project over the conventional nanofluid investment. As such, in all scenarios, the effervescent tablet-based nanofluid project had a lower annual *TAC* compared to the conventional suspension project. Furthermore, the difference in *TAC* between the effervescent tablet-based suspension project and conventional nanofluid investment was found to be highly influenced by the interest rate. The one-time payment case showed the highest difference, from 116.73% (HEC) to 121.39% (LEC), with the balance leaning towards the effervescent tablet production project. This high diversion in *TAC* between the two types of projects reduces with increased interest rate, ranging as low as 37.54% (HEC) to 37.97% (LEC). However, the interest rate on the *TAC* had a much more crucial impact on the effervescent tablet-based project compared to the conventional nanofluid investment. The increase in *TAC* caused by the 30% interest rate over the one-time payment scenario demonstrated a raise of 695.95% (HEC) to 763.88 (LEC) for the effervescent tablet case, whereas the conventional suspension production project showed an increase of 205.98% (HEC) to 210.34% (LEC). Owing to the aforementioned outcomes, investors are likely to opt for effervescent tablet investments if the *TAC* is the primary factor in their consideration between the two projects. Nevertheless, they are likely to evaluate the effects of interest rates in their decisions.

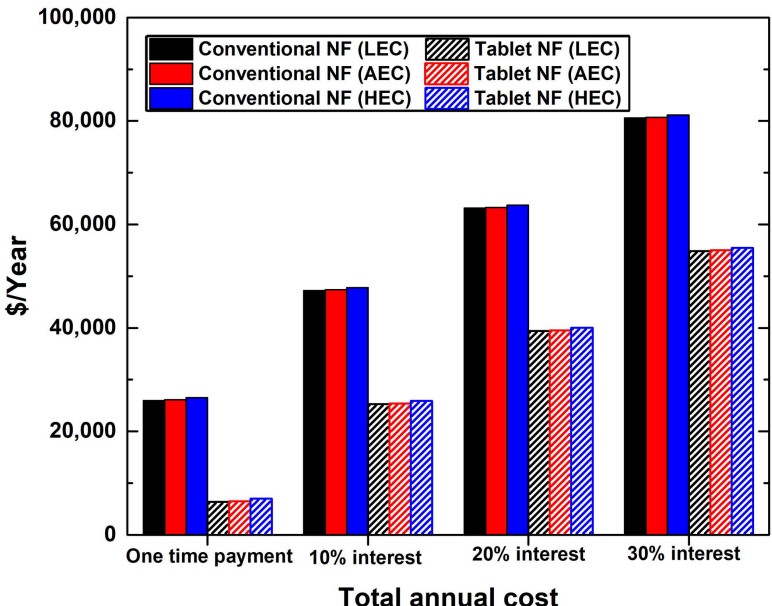

**Fig 7. Changes in total annual cost for the different nanofluid production projects based on the electrical cost and employed interest rate.**

## 3.4. Product cost

The nanofluid production cost was calculated for both production methods while considering the interest rate and electrical cost in the calculation, as shown in Fig 8 (detailed values are available in S7 Table). The *NFPC* was found as 12.70 $/L (lowest) and 162.27 $/L (highest) for the effervescent tablet approach (LEC and one-time payment) and conventional production method (HEC and 30% interest rate) products, respectively. Noticeably, the production cost of the effervescent tablet-based nanofluids is cheaper by 39.09 $/L (HEC) – 39.23 $/L (LEC) and 51.29 $/L (HEC) - 51.43 $/L (LEC) compared to those produced via the conventional process when one-time payment and 30% interest rate are employed, respectively. Moreover, the results indicated a substantial impact of the applied interest rate on the obtained *NFPC* values. This effect is evident in both production scenarios. Regarding nanofluid products derived from the tablet process, the cost increased from 12.70 $/L (LEC) to 13.94 $/L (HEC) as a one-time payment to 109.74 $/L (LEC) – 110.98 $/L (HEC) with a 30% interest rate. Similarly, conventionally fabricated products witnessed an increase in cost from 51.93 $/L (LEC) to 53.03 $/L (HEC) as a one-time payment, and from 161.17 $/L (LEC) to 162.27 $/L (HEC) with a 30% interest rate. Furthermore, based on the calculation results, the electrical cost, as a factor, has a minimal impact on the final nanofluid production cost at a fixed interest rate. It should also be noted that the calculated *NFPC* reflects the actual cost employed by the manufacturer, and not the product-selling cost, which is usually set by the investor.

## 3.5. Project payback and IRR

The payback period is a metric used to assess the time required for a new project to recover its initial investment [57]. It also aids decision makers in evaluating project viability. However, because the value of today's financial assets differs from that in the future (e.g., after 20 years), a discounted payback period analysis is used to obtain a more accurate measure of project profitability. Thus, it offers a tool to estimate the present value of future cash flows in a project

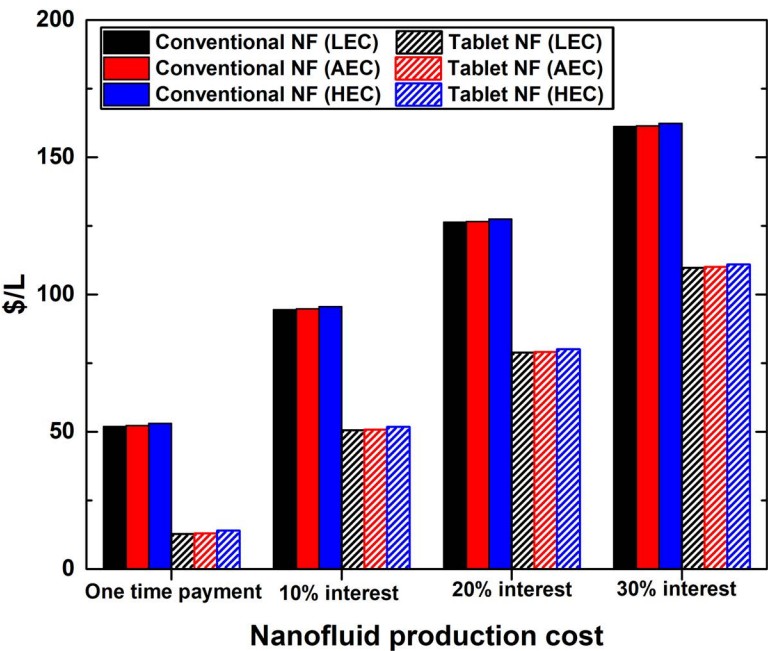

**Fig 8. Nanofluid production cost based on the method of production, electrical cost, and interest rate.**

by considering the impact of discount rates. Conversely, the IRR is another crucial metric for evaluating the profitability of a project [58]. In general, the higher the IRR value, the more attractive a project is to investors and vice versa. However, if a project is found to have a negative IRR value, then the returns from the investment are insufficient for that specific project to be considered financially viable. Hence, losses are anticipated and it would be wiser to consider alternative options. For the previous reasons, the payback period, 10% discounted payback period, and IRR were calculated as shown in Figs 9 and S1 and S8 Table, respectively. Detailed information on Figs 9 and S1 are presented in S9 and S10 Tables, respectively. As shown in Fig 9, when the project production capacity of nanofluids was in the range of 500 L/year and 50 L/year, the resulting payback periods were less than 1.5 years and 8 years, respectively. Specifically, the effervescent tablet-based nanofluids would be in the range of ~ 0.2 years (LEC, no interest rate, 500 L/year) to ~ 6.5 years (HEC, 30% interest rate, 50 L/year), whereas those produced conventionally would have a range of ~ 0.2 (LEC, no interest rate, 500 L/year) to ~ 11.1 years (HEC, 30% interest rate, 50 L/year). Nevertheless, when only 10 L/year of nanofluids was produced, only the effervescent tablet production approach did not exceed the project targeted duration (i.e., 20 years) for any of the three electrical cost scenarios, given that no interest rate was implemented on the investment. Specifically, the previous case illustrated a payback period from ~ 7.7 years (LEC) to ~ 19.7 years (HEC). In addition, when employing a 30% interest rate on the investment, both types of projects exceeded the 20 years cap, except for one case—the effervescent tablet-based project that had an LEC. Moreover, when employing a 10% discount rate in the payback period (S1 Fig), the calculation results demonstrated that the lowest payback period for the effervescent tablet case would be 0.2 years (LEC, no interest rate, 500 L/year) and 15.6 years (LEC, no interest rate, 10 L/year) for the highest that does not exceed the targeted duration of 20 years. As for the conventional production scenario, the lowest payback period would be 0.2 years (LEC, no interest rate, 500 L/year) and 18 years (HEC, 20% interest rate, 50 L/year) for the highest that is within the project duration. It

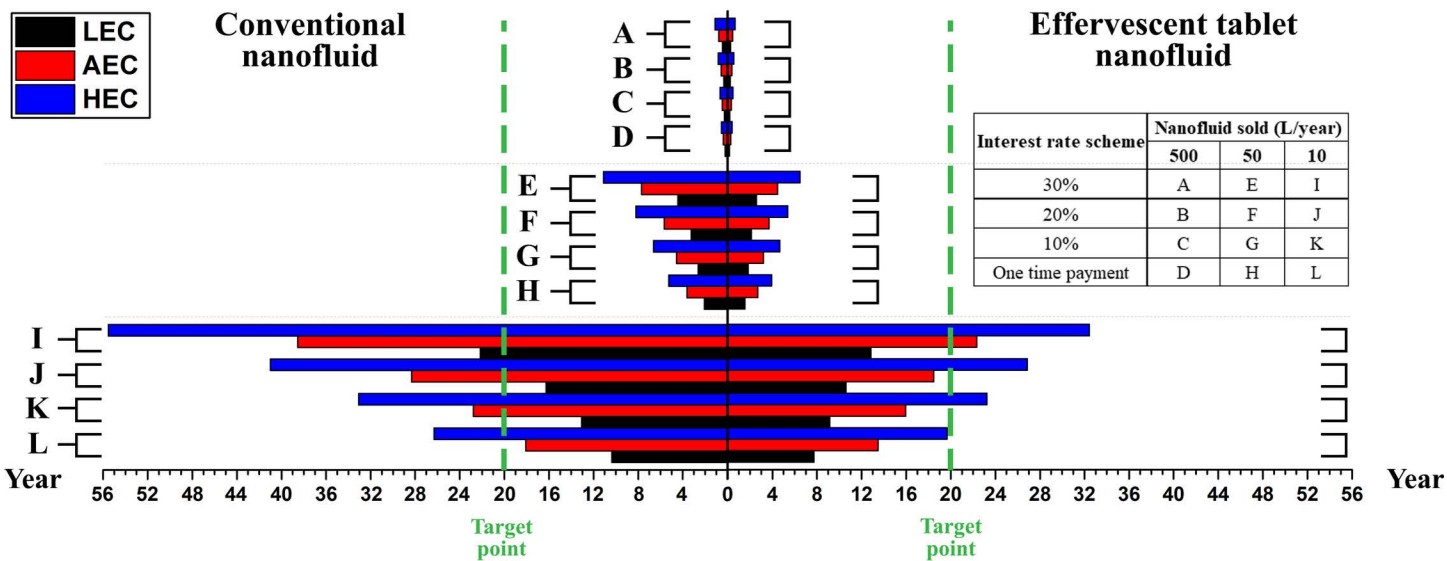

**Fig 9. Project payback period based on the nanofluid method of production, electrical cost, interest rate, and product quantity per year.**

was also found that, regardless of the employed interest rates or electrical cost, both types of projects when selling 10 L/year do not provide a discounted payback period, except for two effervescent tablet-based investments. The first is the LEC and one-time payment case (15.6 years), and the second case is the LEC with 10% interest rate (25.8 years). Although the second case was shown to have a discounted payback period, it failed to maintain the project's targeted duration of 20 years, and thus could not reach the investment goals. In general, project scenarios that show no payback periods have a high tendency to fail; therefore, they should not be executed. As such, investors would refrain from pursuing a project with no discounted payback period or that surpasses the designated duration, such as the restricted 20 years set in this research. However, this can be overcome by adjusting the selling price of the product to support its production costs. This is achieved by setting the $NCF_t$ in Eq. 16 to zero and thus determine the appropriate nanofluid selling price for each $TAC$, interest rate, and OPEX.

The IRR for the different nanofluid fabrication methods, interest rates, electrical rates, and suspension production quantities were determined, as shown in S8 Table. Based on the IRR outcomes, Case 1 (i.e., 500 L/year) for both conventional and effervescent tablet-based nanofluids is favourable from a financial perspective, regardless of the implemented interest rate and electrical cost. This was evident from the positive IRR values for both suspension production approaches. However, in Case 2 (i.e., 50 L/year), the conventional route provided a positive IRR value only when a one-time payment (or 0% interest rate) was adopted for the investment, whereas the IRR for the effervescent tablet-based products was shown to stand out for most of its interest rates scenarios, except for the 30%. For Case 3, where the production is 10 L/year, the investment evidently incurs losses regardless of the fabrication methods, interest rates, or electrical cost, as indicated by the negative return values. Comparing all project scenarios, the effervescent tablet project in Case 1, featuring a one-time payment and HEC, had the highest IRR (i.e., 76.6%/year). As such, it signifies the best financial returns in terms of IRR. In general, the IRR is inversely proportional to the interest rate, showing its lowest value at i = 30% among all three studied cases for both nanofluid fabrication methods. According to Eqs. 9 and 16, increasing the interest rate would result in a higher $TAC$, and consequently leading to a lower net cash flow. The previous causes the IRR value to decrease, according to

Eq. 15. However, an ideal project scenario would usually consist of an IRR value that is higher than that of the employed investment interest rate, as emphasised by Sullivan et al. [59]. Considering the previous criterion, both Cases 2 and 3 are not viable project options. This makes Case 1 the optimal investment choice for both decision makers as well as the investors, with the exception of the 30% interest rate conventional production approach scenario.

### 3.6.  Uncertainty outcome

The effects at the highest and lowest TIC error % on the nanofluid production cost for 0%, 10%, 20% and 30% interest rate are illustrated in Fig 10. The highest deviations in the production cost from the original value for the effervescent tablet and conventional methods were 17.7% and 13.6%, respectively. These values were determined at ± 20% error in the TIC along with 30% interest rate and LEC for both production processes. For a one-time payment, the annual value of the investment cost was zero. Hence, the total annual value depends only on

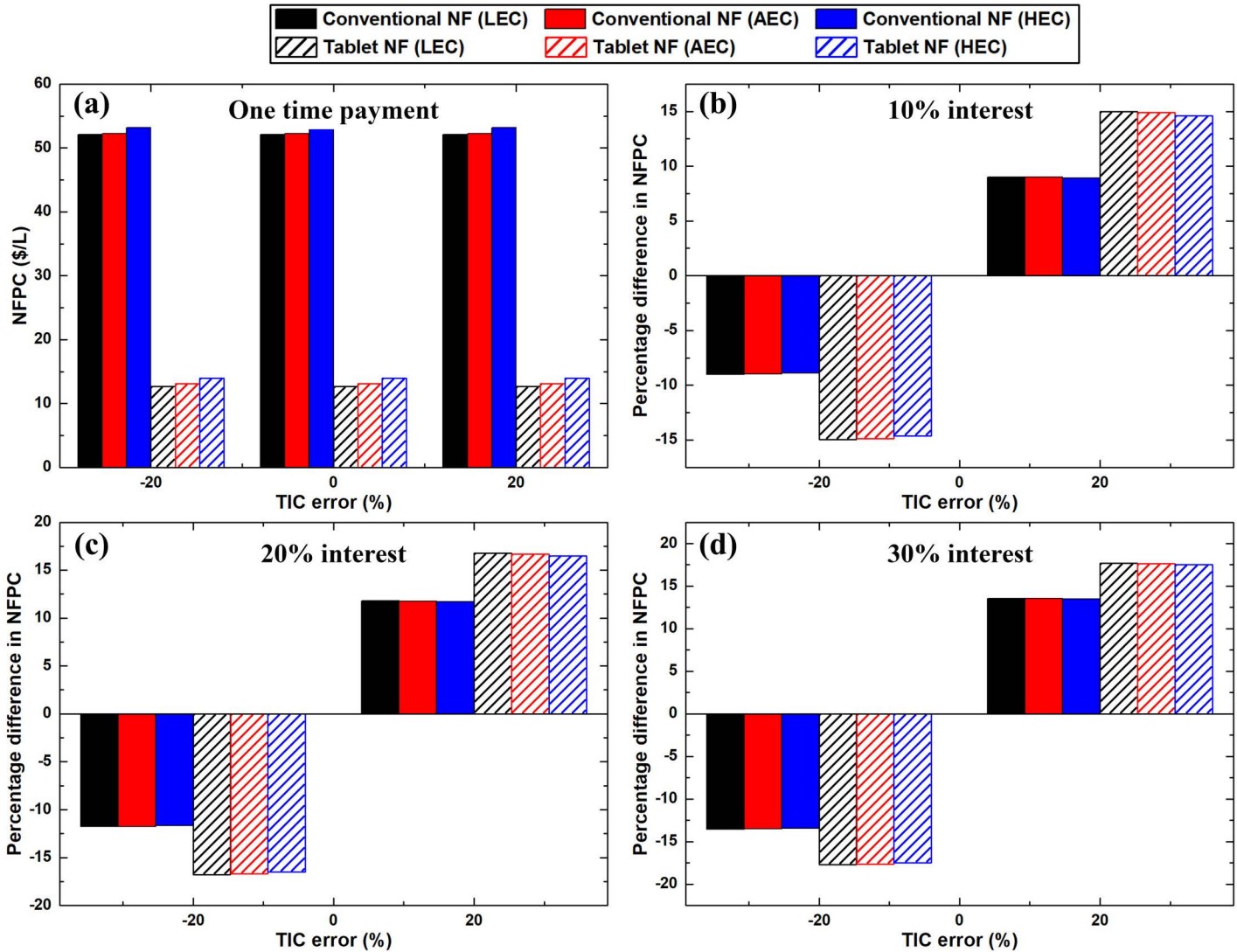

**Fig 10.  Effect of TIC error percentage on the NFPC when employed (a) one-time payment, and (b– d) 10%, 20%, and 30% interest rates, respectively.**

the OPEX according to Eq. 8. Therefore, for both suspension production methods, the TIC error did not affect the total production cost, as shown in Fig 10a.

Furthermore, the impacts of the greatest and least TIC errors % on the IRR values are shown in Fig 11. As shown in Fig 11a–d, the sensitivity of the IRR due to a negative TIC error % was demonstrated as more significant than that due to a positive error %. This is primarily because of the nonlinear nature of Eq. 15, which was used to calculate the IRR. For the effervescent tablet nanofluid production process, the maximum deviation in the IRR for a -20% error in TIC was found to be 41.6%. The aforementioned value was calculated at 30% interest rate with an average electricity price.

A maximum change that is equivalent to 27.9% from the original IRR was obtained for the 20% error in the TIC when 30% interest and LEC were employed. The minimum influence of TIC errors % on the IRR were 16.6% (HEC) and 24.9% (LEC) for + 20% and -20% deviation, respectively. Both values were determined at 0% interest rate (i.e., one-time payment) for the TIC. For the conventional method of nanofluid fabrication, the highest effects of the positive

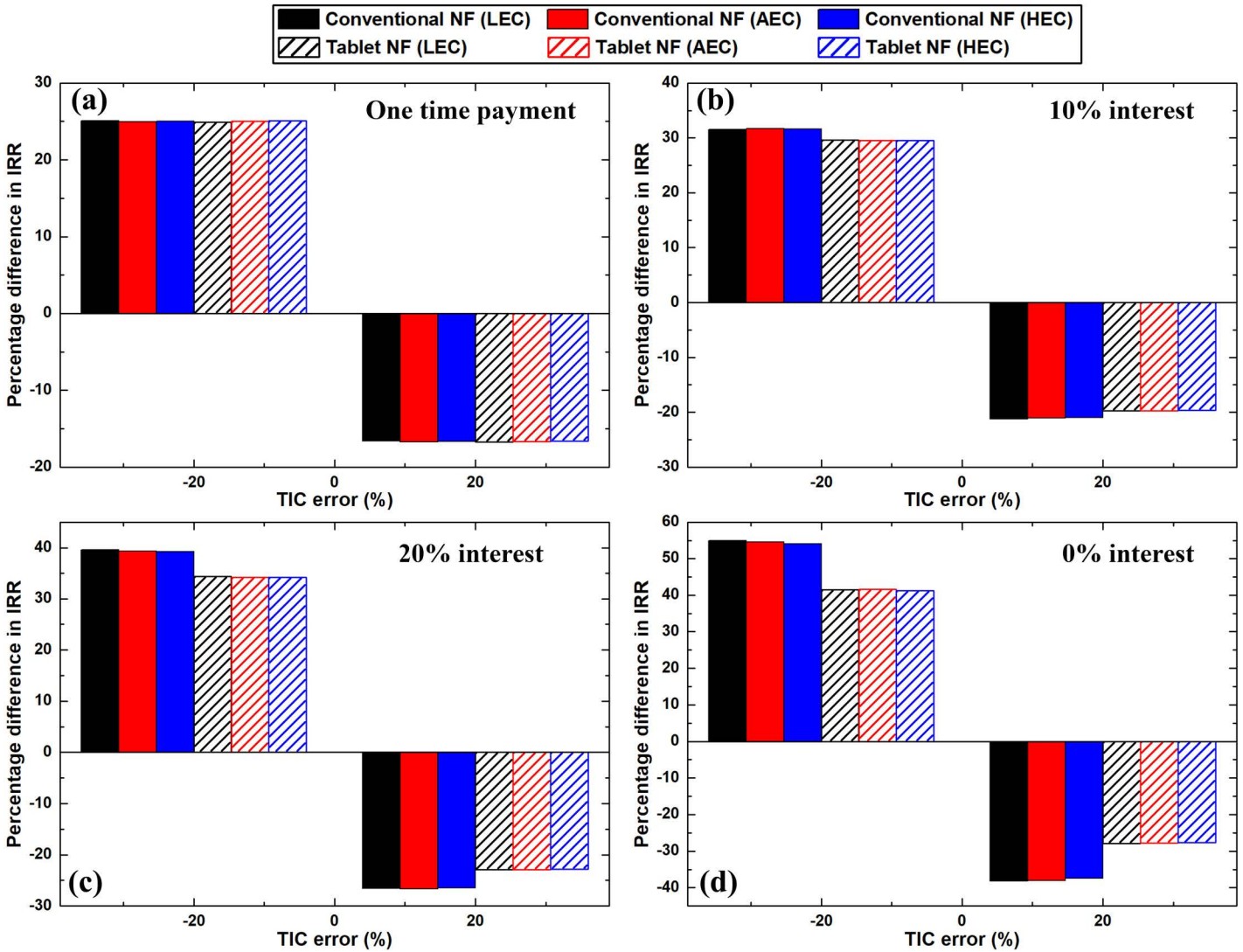

**Fig 11. Effect of TIC error percentage on the IRR when employed (a) one-time payment, and (b– d) 10%, 20%, and 30% interest rates, respectively.**

and negative 20 error % in the TIC on IRR showed to be 38.2% and 55%, respectively, at 30% interest rate with a low electricity price. However, the lowest impact of ± 20% error in TIC was found to be 16.6% (LEC) and 25% (AEC), respectively. The previous values were obtained for one TIC payment (i.e., 0% interest rate).

Conversely, the sensitivity of electricity cost deviation to the nanofluid production cost and the IRR for the two processes were analysed, as illustrated in Figs 12 and 13. Based on Fig 12, the largest effect of the error in electricity cost on the effervescent tablet nanofluid production cost was 1.8%. The previous value was obtained for the ± 20% error with the high electricity price and one-time payment. For the conventional method of nanofluid production, the highest deviation in the fabrication cost from the original value was found as 0.7%. Further, this deviation in the production cost was determined with + 20% error in the high electricity cost with one-time payment.

The influence of the variation in electricity cost on the IRR is shown in Fig 13. The highest variation in the IRR from the original value for the effervescent tablet nanofluid production

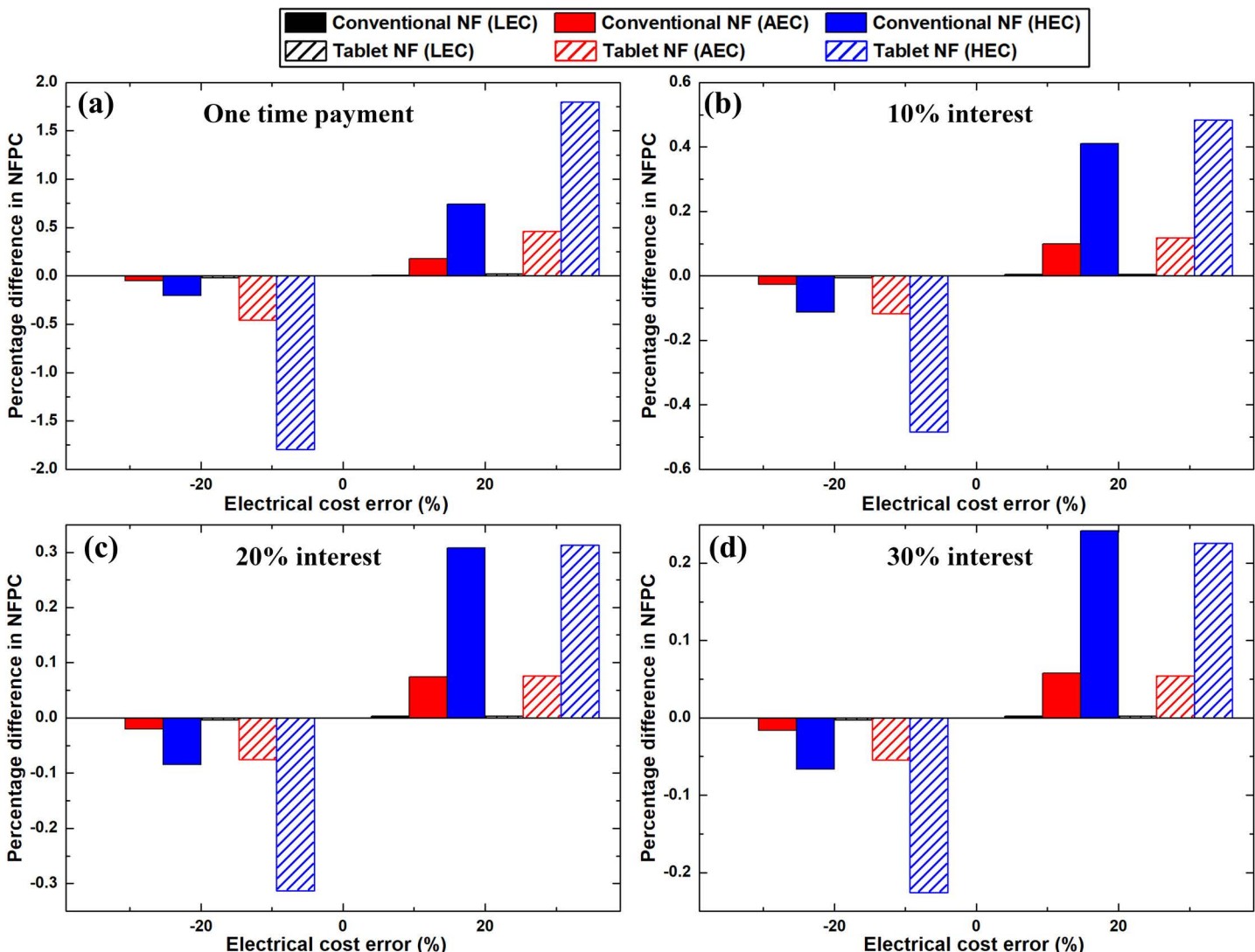

**Fig 12. Effect of electricity cost error percentage on the fabrication cost of the nanofluids when employed (a) one-time payment, and (b–d) 10%, 20%, and 30% interest rates, respectively.**

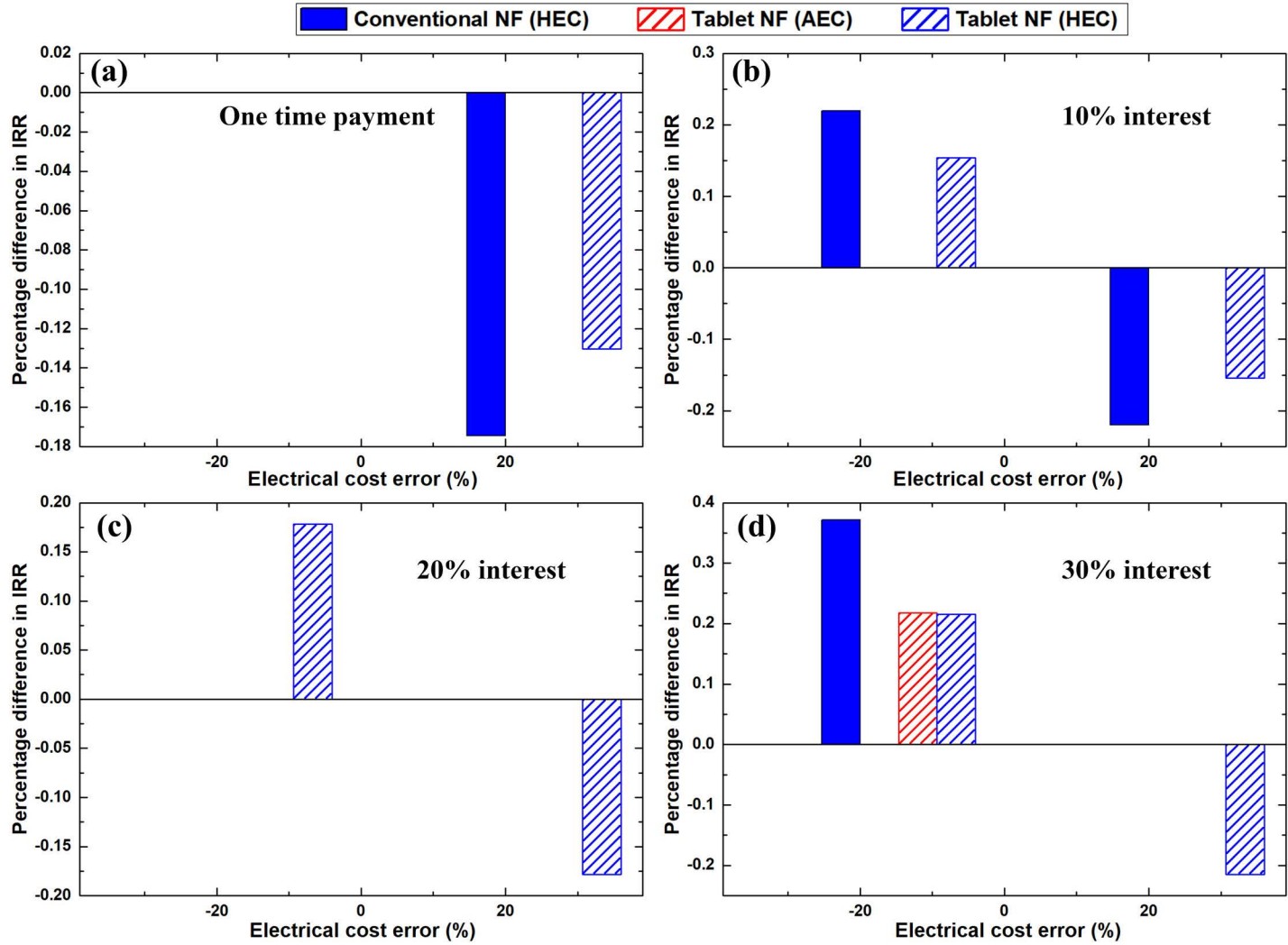

**Fig 13. Effect of electricity cost error percentage on the IRR when employed (a) one-time payment, and (b–d) 10%, 20%, and 30% interest rates, respectively.**

process was 0.2%, which was obtained with a 30% interest rate and AEC. Regarding the conventional nanofluids, the highest variation in the IRR from its original value was 0.4% (HEC), which was obtained at the highest negative error margin with 30% interest rate. Based on these values, the impact of electricity cost errors on the nanofluid production cost and IRR was insignificant. This is attributed to the small share of electricity cost from OPEX. Specifically, the electricity cost comprises only 0.1% – 8.9% of the total OPEX for the effervescent tablet nanofluid production process for all three electricity prices. Regarding the conventional method, the OPEX comprises only 0.02% – 2.1% electricity cost for all three-electricity prices, while the remainder is mainly the consumables cost. This indicates that the percentage difference between the IRR at the highest positive and negative deviations in electricity cost is approximately zero.

### 3.7. Thermal performance and price-performance factor

Fig 14a–c demonstrates the values of $C_\mu/C_\lambda$, Mo, and $FOM_{Heat\ losses}$, respectively for the conventional and effervescent tablet-based nanofluids as a function of temperature. Based on Fig 14a,

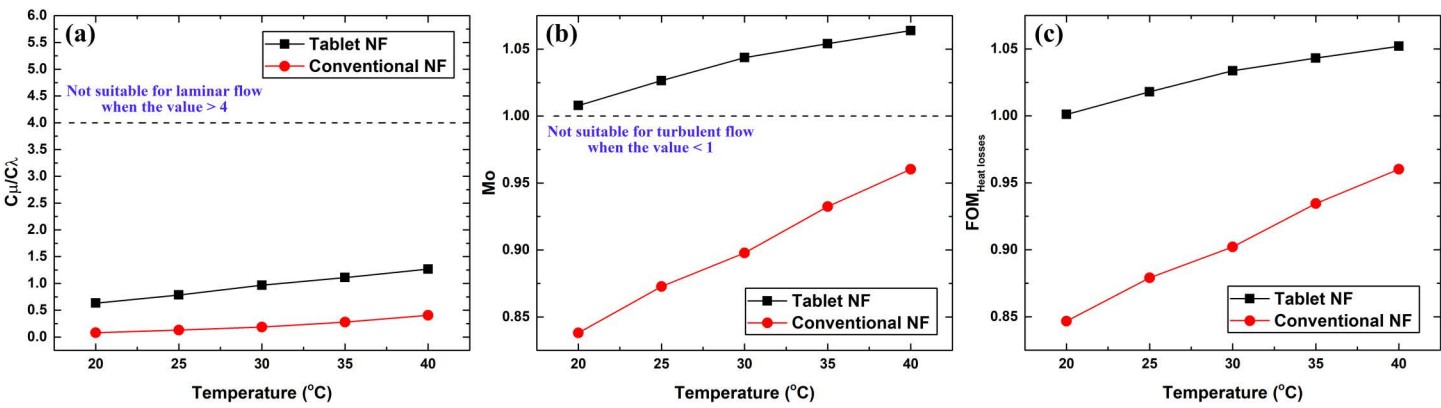

**Fig 14. Variation of (a) $C_\mu/C_\lambda$, (b) Mo, and (c) FOM$_{Heat\ losses}$ with respect to temperature and nanofluid type.**

the $C_\mu/C_\lambda$ values of both types of nanofluids are all below 4, which indicates that the rise in thermal conductivity is higher than the enhancement in viscosity. Thus, both types of nanofluids are suitable for use in the laminar flow regime. However, the outcome of Mo (Fig 14b) demonstrates that only the effervescent tablet-based suspension is appropriate for use under turbulent flow conditions, as all its values are above 1. Regarding the FOM$_{Heat\ losses}$, the effervescent tablet-based nanofluid exhibited a better overall heat transfer performance compared to its counterpart; as such, it is preferable for thermal applications. The FOM$_{Heat\ losses}$ values obtained for the effervescent tablet-based nanofluid is higher than the conventional dispersion by 18.24% and 9.56% at 20 °C and 40 °C, respectively.

Figs 15 and 16 shows the variation in PPF$_C$ and PPF$_{Mo}$ as a function of interest rate, electrical cost, temperature, and nanofluid type. Based on Fig 15, the electrical cost is an insignificant factor influencing PPF$_C$. Further, the effervescent tablet-based nanofluid has a higher economic feasibility under laminar flow condition over the conventional counterpart at 40 °C and one-time payment, as shown in Fig 15a. Furthermore, the conventional suspension generally showed a higher PPF$_C$ than its counterpart, regardless of the interest rate applied. This indicates that conventional nanofluids are more cost-effective under laminar flow conditions. Conversely, under turbulent flow conditions, the effervescent tablet-based nanofluid exhibits higher PPF$_{Mo}$ values than the conventional suspension, as shown in Fig 16. In addition, the electrical cost was shown to be effective only for one-time payments. In general, it can be concluded that the effervescent tablet-based nanofluid has better cost-effectiveness under turbulent flow conditions within the different factors considered.

## 4. Conclusion

This study analyses the techno-economics of two start-up industrial projects that deal with the production and sale of nanofluids. These nanofluids are favourable to many thermal related industries owing to their considerable thermal performance and operational enhancement capability. Two routes of nanofluid fabrication were considered in this study: the conventional two-step method and the effervescent tablet-based approach. The investigation assessed the thermophysical properties of the nanofluids across 20 °C to 40 °C, alongside cost analyses that included equipment, consumables, operational expenditures, and electrical cost of running the devices on three worldwide scales (LEC, AEC, HEC). Additionally, capital costs were analyzed under various payment scenarios with 10%–30% interest rates over a 20-year project duration.

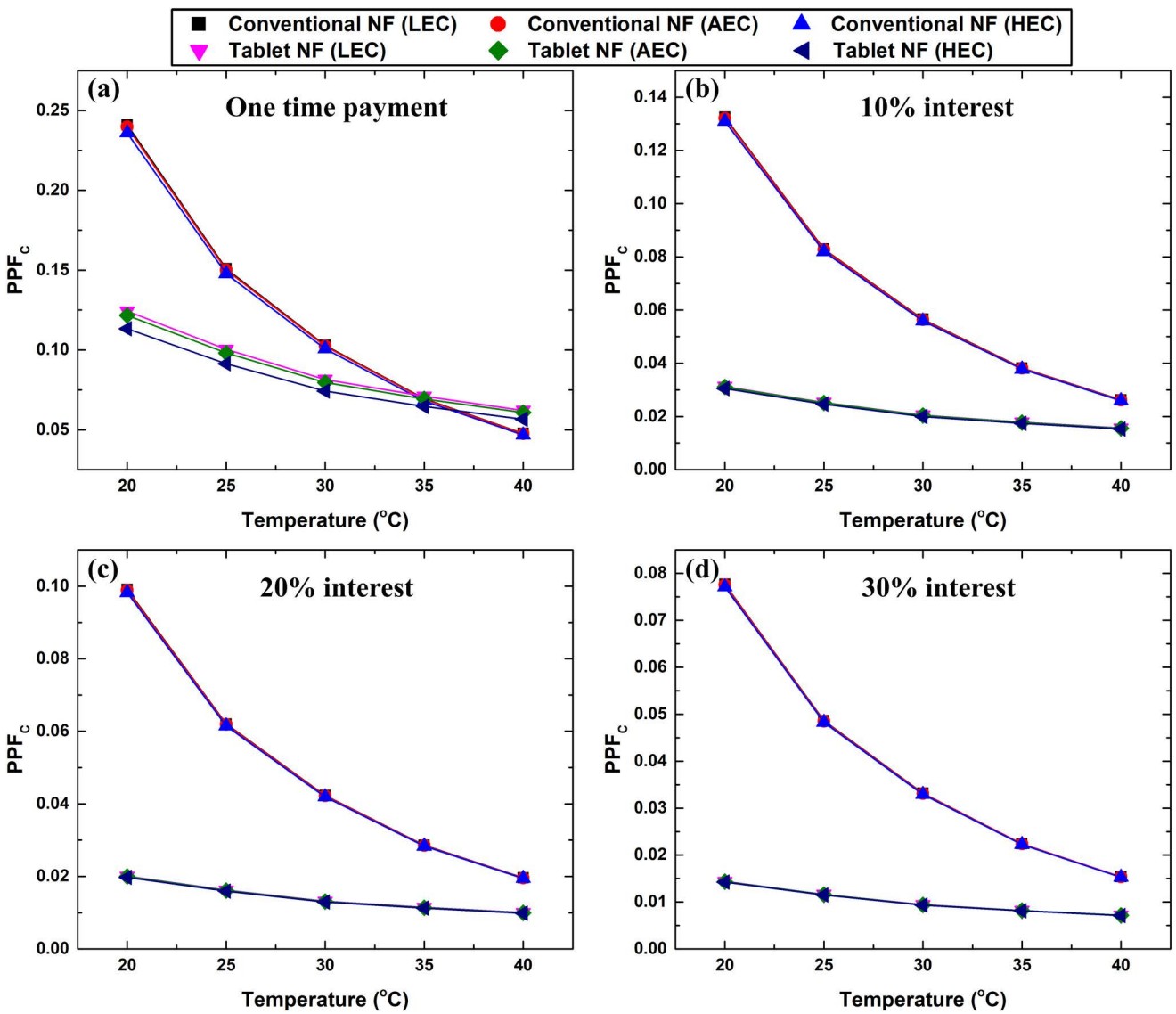

**Fig 15.** PPF$_C$ variation with temperature and electrical cost for the conventional and effervescent tablet-based nanofluids when considering (a) one-time payment, (b–d) 10%, 20%, and 30% interest rates, respectively.

Both types of nanofluids exhibit improved thermophysical properties when compared to their basefluid, with the effervescent tablet-based nanofluids over performing their counterparts. For the cost analysis, the equipment cost is the primarily parameter affecting the investments. A 13.26% difference in the equipment cost between the two types of projects favors the effervescent tablet-based project, resulting in it being cheaper by 12.57% than the conventional nanofluid projects. On the other hand, implementing 10% to 30% interest rates over the 20-year period significantly impacted the total projects cost. Thus, leading to an increase in the cost of the effervescent tablet project and the conventional nanofluid project from $160,883.39 to $970,406.43 and $181,104.3 to $1,092,370, respectively. On the contrary, investment settlement strategies demonstrate high saving capability towards the projects. Accordingly, showing savings in the range of 16.05% to 43.85% (10% interest) and 23.91% to 65.97% (30% interest) for both projects, depending on the selected settling year.

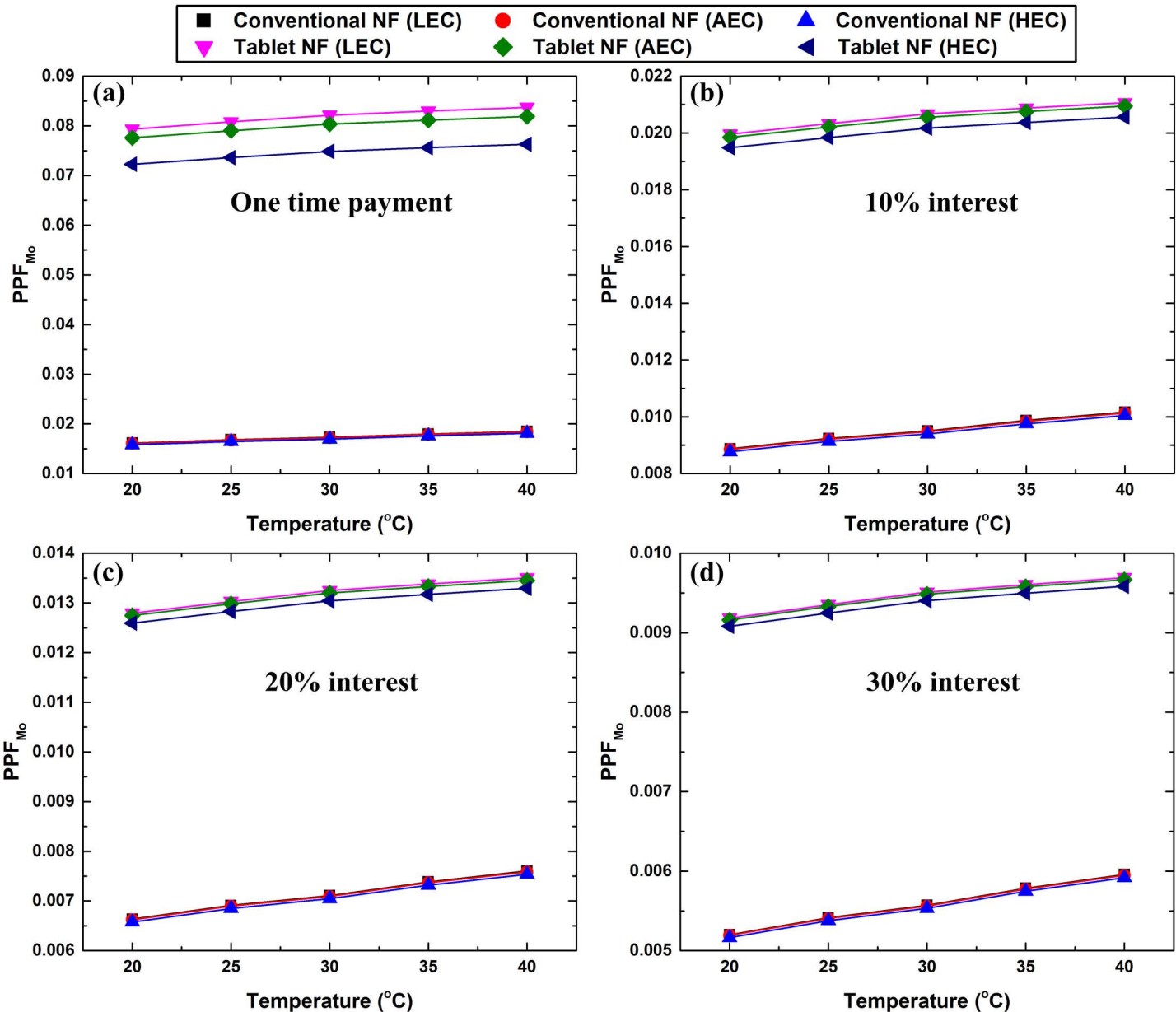

**Fig 16.** $PPF_{Mo}$ variation with temperature and electrical cost for the conventional and effervescent tablet-based nanofluids when considering (a) one-time payment, (b–d) 10%, 20%, and 30% interest rates, respectively.

In terms of the projects OPEX and TAC, the effervescent tablet project consistently exhibited lower OPEX (116.73%-121.39%) and yearly TAC than the conventional suspension project; thus, underscoring the favourability of the tablet-based project over its counterpart. Evaluation of the production costs showed that the effervescent tablet project was cheaper than the conventional nanofluid project by a range of ~ 39 \$/L - ~ 51 \$/L (zero to 30% interest). Project payback period, discounted payback period, and IRR results indicates that both projects could be successful if 500 L/year of nanofluids were sold, regardless of the implemented type of investment, whereas the balance favoured the effervescent tablet-based project. Effervescent tablet-based nanofluid exhibits superior thermal performance than its

counterpart when used in both laminar and turbulent flow conditions. However, its cost effectiveness was found to be only under turbulent flow conditions, making such type of suspensions beneficial for selective applications.

In conclusion, the effervescent tablet-based nanofluid projects offers a more favourable investment opportunity for decision-makers. Future studies should expand to include a broader variety of nanomaterials and assess consumer preferences in diverse geographical contexts to refine the results and enhance industrial impact.

## Nomenclature

| | |
|---|---|
| $A_c$ | Annual value of the total investment cost |
| AEC | Average electrical cost |
| FOM | Figure of merit |
| HEC | Highest electrical cost |
| $i$ | Interest rate percentage |
| IRR | Internal rate of return |
| LEC | Lowest electrical cost |
| Mo | Mouromtseff number |
| MWCNTs | Multi-walled carbon nanotubes |
| $n$ | Lifetime of the project |
| $Na_2CO_3$ | Sodium carbonate |
| $NaH_2PO_4$ | Monosodium phosphate |
| NF | Nanofluid |
| $NF_p$ | Nanofluid annual production rate |
| NFPC | Nanofluid production cost |
| $NFPC_{avg}$ | Average nanofluid production cost |
| OPEX | Operating expenses |
| PCM | Phase change material |
| PPF | Price-performance factor |
| PV | Photovoltaic |
| SDS | Sodium dodecyl sulfate |
| T | Total number of time periods |
| t | Time period |
| TAC | Total annual cost |
| $\lambda$ | Thermal conductivity (W/m.K) |
| THW | Transient hot wire |
| TIC | Total investment cost |
| $\mu$ | Viscosity (mPa.s) |

## Supporting information

**S1 Table. Prices of the equipment, tools, and part used to produce the different nanofluids.**
(PDF)

**S2 Table. Prices of powders used to produce the different nanofluids.**
(PDF)

**S3 Table. Amount of electricity required to run the devices and its cost.**
(PDF)

**S4 Table. Accumulated cost for the conventional two-step and effervescent tablet-based nanofluid production project at different interest rates.**
(PDF)

**S5 Table. Accumulated interest cost for the conventional two-step and effervescent tablet-based nanofluid production project at different set of interest rates.**
(PDF)

**S6 Table. Changes in total annual cost for the different nanofluid production projects based on the electrical cost and employed interest rate.**
(PDF)

**S7 Table. Nanofluid production cost based on the method of production, electrical cost, and interest rate.**
(PDF)

**S8 Table. Project internal rate of return based on the nanofluid method of production, electrical cost, interest rate, and product quantity per year.**
(PDF)

**S9 Table. Detailed information on the payback period for conventional and effervescent tablet-based nanofluid projects.**
(PDF)

**S10 Table. Detailed information on the 10% discounted payback period for conventional and effervescent tablet-based nanofluid projects.**
(PDF)

**S1 Fig. Project 10% discounted payback period based on the nanofluid method of production, electrical cost, interest rate, and product quantity per year.**
(TIF)

## Acknowledgment

The authors acknowledge the Kuwait Foundation for the Advancement of Sciences (KFAS), Kuwait Institute for Scientific Research (KISR), Kuwait University (KU), and Public Authority for Applied Education and Training (PAAET) for their help and support during this study.

## Author contributions

**Conceptualization:** Naser Ali, Husain Bahzad.

**Data curation:** Naser Ali.

**Formal analysis:** Naser Ali, Husain Bahzad.

**Funding acquisition:** Naser Ali.

**Investigation:** Naser Ali, Husain Bahzad, Nawaf F. Aljuwayhel, Shikha A. Ebrahim.

**Methodology:** Naser Ali, Nawaf F. Aljuwayhel.

**Project administration:** Naser Ali.

**Resources:** Naser Ali, Husain Bahzad, Nawaf F. Aljuwayhel, Shikha A. Ebrahim.

**Supervision:** Naser Ali.

**Visualization:** Naser Ali, Nawaf F. Aljuwayhel, Shikha A. Ebrahim.

**Writing – original draft:** Naser Ali.

**Writing – review & editing:** Naser Ali, Husain Bahzad, Nawaf F. Aljuwayhel, Shikha A. Ebrahim.

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
