## [Decision Letter · Decision Letter 0]

8 Sep 2024

PONE-D-24-29775Techno-Economic Assessment of Effervescent Tablet-Based NanofluidsPLOS ONE

Dear Dr. Ali,

Thank you for submitting your manuscript to PLOS ONE. After careful consideration, we feel that it has merit but does not fully meet PLOS ONE’s publication criteria as it currently stands. Therefore, we invite you to submit a revised version of the manuscript that addresses the points raised during the review process.

**Address all comments following the reviewers concerns. In addition, there are issues with inconsistent referencing, grammatical errors and structure of the manuscript. Images should be clear. All codes should be provided. **

We look forward to receiving your revised manuscript.

Kind regards,

Jude Okolie, Ph.D.

Academic Editor

PLOS ONE

“The authors acknowledge the financial and technical support that were provided by the funding agency, KFAS (grant no: PN23-35EM-1837), and their institutes, KISR (grant no: EA112C) to complete this research work.”

“N.A. received fund from the Kuwait Foundation for the Advancement of Sciences (KFAS) under project grant no: PN23-35EM-1837. KFAS website: www.kfas.org

The funder did not play any role in the study design, data collection and analysis, decision to publish, or preparation of the manuscript.”

3. We note that Figures 1 and 2 in your submission contain copyrighted images. All PLOS content is published under the Creative Commons Attribution License (CC BY 4.0), which means that the manuscript, images, and Supporting Information files will be freely available online, and any third party is permitted to access, download, copy, distribute, and use these materials in any way, even commercially, with proper attribution. For more information, see our copyright guidelines: http://journals.plos.org/plosone/s/licenses-and-copyright.

1. You may seek permission from the original copyright holder of Figures 1 and 2 to publish the content specifically under the CC BY 4.0 license.

4. Please remove your figures from within your manuscript file, leaving only the individual TIFF/EPS image files, uploaded separately. These will be automatically included in the reviewers’ PDF.

Reviewers' comments:

Reviewer's Responses to Questions

**Comments to the Author**

1. Is the manuscript technically sound, and do the data support the conclusions?

Reviewer #1: Partly

Reviewer #2: Yes

2. Has the statistical analysis been performed appropriately and rigorously? 

Reviewer #1: Yes

Reviewer #2: No

3. Have the authors made all data underlying the findings in their manuscript fully available?

Reviewer #1: Yes

Reviewer #2: Yes

4. Is the manuscript presented in an intelligible fashion and written in standard English?

Reviewer #1: No

Reviewer #2: Yes

5. Review Comments to the Author

Reviewer #1: The novelty of this paper is evident, but it is unfortunately difficult to read due to numerous grammatical and syntactical errors. Some sentences are overly complex and require simplification for clarity and conciseness.

Additionally, the introduction covers several topics—energy demand, nanotechnology, nanofluids, and methods of fabrication—without a clear central theme. It is also excessively lengthy and repetitive.

In the research methods section, the correspondence between terms and variables in the equations is unclear.

Figure 1 appears in both this document and the article “Effervescent Tablets for Carbon-Based Nanofluids Production” on ScienceDirect. While it is cited, it is not clear that the figure is sourced from that paper. The same issue applies to Figure 2.

Equations 12-14 could be improved by adding more acronyms to enhance readability and conciseness.

In Figures 11 and 12, the legend includes identifiers that do not match those in the graphs. Additionally, Figure 14 is difficult to read.

In the conclusion, the presentation of information and data needs to be clearer and more concise.

If the journal is to accept this paper, significant editing and restructuring of the presentation will be necessary to address these issues and improve overall readability.

Reviewer #2: The authors attempt the (novel) techno-economic assessment of effervescent tablet-based nanofluids, through the two-step synthesis approach to compare both the conventional and tablet-based nanofluids. The study assessed several financial metrics based on the two case studies, and the result confirmed the superiority of the tablet-based nanofluid over the conventional approach. This is indeed an important study, as these nanofluids are suitable as alternative working fluids in thermal systems, and the authors have done a great detailed study from an economic standpoint. Although in its early stage, tablet-based nanofluids appear to be a viable alternative, owing to their low OPEX, higher thermal performance, and the amenability to work in both laminar and turbulent flow conditions. Indeed, the manuscript provides sufficient details regarding the techno-economic study, and it is favorable to proceed into the next phase. However, the following questions/concerns need to be addressed before the manuscript proceeds to the next phase:

1. On page 18, replace the statement ‘For instant…’ with ‘For instance,’ the author should review most of the result and discussion section for any grammatical error.

2. Statistically speaking, Fig. 6 shows no substantial differences in total OPEX values among the scenarios examined. This raises the question of whether the different nanofluid production methods, based on electrical costs, have a minimal effect on OPEX for both tablet and conventional production. The authors may want to provide further clarification, given the narrow range of values.

3. Would it be possible to tabulate the main economic indicators/metrics or values from sections 3.2 to 3.4 for easier comprehension? The current display of data is extensive and may obscure the key points.

4. Fig. 9 and 10, seem to be quite a complex figure with many details that may throw the reader off. If the authors can create a table for the keys/legend, that clearly describes each alphabet from A to L. If the authors can generate a table that simplifies the results from the plot, it would be clear to the readers.

5. A lot of graphs/plots and tables shown can be moved to the supplementary information (SI). For example, Table 6 can be sent to SI.

6. From the TEA study, the authors proposed that the tablet-based nanofluid is more economical compared to the conventional approach. Is it possible for the authors to comment on the minimum selling price of these prepared nanofluids to support the production cost?

6. PLOS authors have the option to publish the peer review history of their article (what does this mean?). If published, this will include your full peer review and any attached files.

Reviewer #1: **Yes: **Brooke Rogachuk

Reviewer #2: No

---

## [Author Response · Author response to Decision Letter 1]

8 Oct 2024

We would like to start by thanking the respected Academic Editor, respected Reviewer 1, and respected Reviewer 2 for taking the time to review our manuscript and for their variable comments and suggestions. Kindly find below our reply on these comments.

Academic Editor

Academic Editor comment:

Authors reply:

We thank the respected Academic Editor for his time and for reviewing our manuscript. Kindly note that the revised version of the manuscript was modified according to PLOS ONE’s style requirements, including those for file naming, as instructed by the respected Academic Editor. Thank you very much.

Academic Editor comment:

“The authors acknowledge the financial and technical support that were provided by the funding agency, KFAS (grant no: PN23-35EM-1837), and their institutes, KISR (grant no: EA112C) to complete this research work.”

Please remove any funding-related text from the manuscript and let us know how you would like to update your Funding Statement.

Authors reply:

All funding related information was removed from the revised version of the manuscript, as instructed by the respected Academic Editor. We have also written and updated funding statement in our cover latter based on the instructions provided by the respected Academic Editor, which is as following:

The authors acknowledge the financial and technical support that were provided by the Kuwait Foundation for the Advancement of Sciences (KFAS) under project grant no: PN23-35EM-1837, and the Kuwait Institute for Scientific Research (KISR) under project grant no: EA112C to complete this research work. KFAS website: www.kfas.org; and KISR website: www.kisr.edu.kw

Thank you very much.

Academic Editor comment:

3. We note that Figures 1 and 2 in your submission contain copyrighted images. All PLOS content is published under the Creative Commons Attribution License (CC BY 4.0), which means that the manuscript, images, and Supporting Information files will be freely available online, and any third party is permitted to access, download, copy, distribute, and use these materials in any way, even commercially, with proper attribution. For more information, see our copyright guidelines: http://journals.plos.org/plosone/s/licenses-and-copyright.

We require you to either (1) present written permission from the copyright holder to publish these figures specifically under the CC BY 4.0 license, or (2) remove the figures from your submission

Authors reply:

We have replaced Figures 1 and 2 with new figures in our modified version of the manuscript. Please see a comparison between the figures that were referred to in the literature (https://doi.org/10.1016/j.molliq.2023.123083) with the new ones that were included in the modified version of the manuscript below. Thank you very much.

Academic Editor comment:

4. Please remove your figures from within your manuscript file, leaving only the individual TIFF/EPS image files, uploaded separately. These will be automatically included in the reviewers’ PDF.

Authors reply:

All figures have been removed from the revised version of the manuscript as instructed by the respected Academic Editor. Thank you very much.

Academic Editor comment:

Authors reply:

The reference list has been revised and modified in the revised version of the manuscript as instructed by the respected Academic Editor. Thank you very much.

Reviewer 1

Reviewer 1 comment:

The novelty of this paper is evident, but it is unfortunately difficult to read due to numerous grammatical and syntactical errors. Some sentences are overly complex and require simplification for clarity and conciseness.

Authors reply:

We thank the respected reviewer for reviewing our manuscript and for pointing this out. Kindly note that the revised version of the manuscript was sent to a language editing service to improve its readability and revise all grammatical and syntactical errors. Accordingly, all errors were fixed along with the text flow. Kindly check below the language editing certificate. We do apologies for the previous errors included in our initial manuscript. Thank you very much

Reviewer 1 comment:

Additionally, the introduction covers several topics—energy demand, nanotechnology, nanofluids, and methods of fabrication—without a clear central theme. It is also excessively lengthy and repetitive.

Authors reply:

We thank the respected reviewer for the provided comment, which we agree with and were happy to take into account. Kindly note that the Introduction Section was revised to focus on nanofluids. As such, we have modified and shortened it by removing unnecessarily parts. Kindly check Lines 53-123. Thank you very much.

Reviewer 1 comment:

In the research methods section, the correspondence between terms and variables in the equations is unclear.

Authors reply:

We thank the respected reviewer for pointing this out. Kindly note that we have modified our revised version to further clarify the correspondence between terms and variables in the equations, as suggested by the respected reviewer. Kindly check the Research Methodology Sections 2.1, 2.2, 2.3, 2.4, and 2.5. All changes and modifications are in red text. Thank you very much.

Reviewer 1 comment:

Figure 1 appears in both this document and the article “Effervescent Tablets for Carbon-Based Nanofluids Production” on ScienceDirect. While it is cited, it is not clear that the figure is sourced from that paper. The same issue applies to Figure 2.

Authors reply:

The authors have changed both figures in their revised version of the manuscript. Kindly see below a comparison between our two figures and the ones in the provided literature. Thank you very much.

Reviewer 1 comment:

Equations 12-14 could be improved by adding more acronyms to enhance readability and conciseness.

Authors reply:

The authors thank the respected reviewer for the suggestion, which we were happy to take into account in our revised version of the manuscript. Kindly note that more acronyms were included in the pointed-out equations. Kindly check Sections 2.4 and 2.5. Thank you very much.

Reviewer 1 comment:

In Figures 11 and 12, the legend includes identifiers that do not match those in the graphs. Additionally, Figure 14 is difficult to read.

Authors reply:

The authors apologies for the poor illustration. We have changed the illustration of the pointed-out figures to better present the data and eliminate the over lapping. Kindly check new figures 10-13. In addition, please note that the Fig. numbers has changed because we have moved one of the manuscript figures to the Supplementary Information Section. Thank you very much.

Reviewer 1 comment:

In the conclusion, the presentation of information and data needs to be clearer and more concise.

Authors reply:

The authors thank the respected reviewer for the comment. Kindly note that the Conclusion Section was modified to have a clearer presentation of the information, as suggested by the respected reviewer. Thank you very much.

Reviewer 2

Reviewer 2 comment:

On page 18, replace the statement ‘For instant…’ with ‘For instance,’ the author should review most of the result and discussion section for any grammatical error.

Authors reply:

We thank the respected reviewer for reviewing our manuscript and for pointing this out. Kindly note that the revised version of the manuscript was sent to a language editing service to improve its readability and revise all grammatical and syntactical errors. Accordingly, all grammatical errors were fixed along with the text flow. Kindly check below the language editing certificate. Thank you very much and we do apologies for the previous errors included in our initial manuscript.

Reviewer 2 comment:

Statistically speaking, Fig. 6 shows no substantial differences in total OPEX values among the scenarios examined. This raises the question of whether the different nanofluid production methods, based on electrical costs, have a minimal effect on OPEX for both tablet and conventional production. The authors may want to provide further clarification, given the narrow range of values.

Authors reply:

We thank the respected reviewer for his comment. Kindly note that the following statement was added to the revised version of the manuscript (Lines 385-388) for further clarification:

‘Notably, the electrical cost represents 0.02% (LEC) to 2.09% (HEC) and 0.10% (LEC) to 8.98% (HEC) of the total yearly OPEX cost for the conventional and effervescent tablet production methods, respectively. Further information on the values used to generate Fig 6 is presented in S5 Table.’

Thank you very much.

Reviewer 2 comment:

Would it be possible to tabulate the main economic indicators/metrics or values from sections 3.2 to 3.4 for easier comprehension? The current display of data is extensive and may obscure the key points.

Authors reply:

We thank the respected reviewer for his comment. Kindly note that we have tabulated the data from sections 3.2 to 3.4, as suggested by the respected reviewer. Kindly check S4 Table, S5 Table, S6 Table, and S7 Table in the Supplementary Information Section. Thank you very much.

Reviewer 2 comment:

Fig. 9 and 10, seem to be quite a complex figure with many details that may throw the reader off. If the authors can create a table for the keys/legend, that clearly describes each alphabet from A to L. If the authors can generate a table that simplifies the results from the plot, it would be clear to the readers.

Authors reply:

We thank the respected reviewer for his comment. Kindly note that we have included a table in the two pointed-out figures, as suggested by the respected reviewer. Kindly check Fig 9 and S8 Fig. Also, please check S10 Table and S11 Table. Thank you very much.

Reviewer 2 comment:

A lot of graphs/plots and tables shown can be moved to the supplementary information (SI). For example, Table 6 can be sent to SI.

Authors reply:

As suggested by the respected reviewer, we have removed previously Table 6 (new S9 Table) to the SI along with other Tables and Figure. Kindly check in the SI Section: S1 Table, S2 Table, S3 Table, and S8 Fig. Thank you very much.

Reviewer 2 comment:

From the TEA study, the authors proposed that the tablet-based nanofluid is more economical compared to the conventional approach. Is it possible for the authors to comment on the minimum selling price of these prepared nanofluids to support the production cost?

Authors reply:

The authors thank the respected reviewer for his comment. Kindly note that we have commented on the minimum selling price of these prepared nanofluids to support the production cost in our revised version of the manuscript, as following:

‘However, this can be overcome by adjusting the selling price of the product to support its production costs. This is achieved by setting the in Eq. 16 to zero and thus determine the appropriate nanofluid selling price for each TAC, interest rate, and OPEX.’. Thank you very much.

Finally, we would like to thank the respected Academic Editor, respected Reviewer 1, and respected Reviewer 2 for their time, comments, and suggestions, which helped in improving our manuscript.

Thank you all very much.

---

## [Decision Letter · Decision Letter 1]

20 Nov 2024

PONE-D-24-29775R1Techno-economic assessment of effervescent tablet-based nanofluidsPLOS ONE

Dear Dr. Ali, 

Thank you for submitting your manuscript to PLOS ONE. After careful consideration, we feel that it has merit but does not fully meet PLOS ONE’s publication criteria as it currently stands. Therefore, we invite you to submit a revised version of the manuscript that addresses the points raised during the review process.

We look forward to receiving your revised manuscript.

Kind regards,

Jude Okolie, Ph.D.

Academic Editor

PLOS ONE

Journal Requirements:

Reviewers' comments:

Reviewer's Responses to Questions

**Comments to the Author**

1. If the authors have adequately addressed your comments raised in a previous round of review and you feel that this manuscript is now acceptable for publication, you may indicate that here to bypass the “Comments to the Author” section, enter your conflict of interest statement in the “Confidential to Editor” section, and submit your "Accept" recommendation.

Reviewer #1: All comments have been addressed

Reviewer #2: All comments have been addressed

2. Is the manuscript technically sound, and do the data support the conclusions?

Reviewer #1: Yes

Reviewer #2: Yes

3. Has the statistical analysis been performed appropriately and rigorously? 

Reviewer #1: Yes

Reviewer #2: Yes

4. Have the authors made all data underlying the findings in their manuscript fully available?

Reviewer #1: Yes

Reviewer #2: Yes

5. Is the manuscript presented in an intelligible fashion and written in standard English?

Reviewer #1: Yes

Reviewer #2: Yes

6. Review Comments to the Author

Reviewer #1: The authors did a great job with the edits! However, there are still a few issues in the paper. Firstly, the figures are not displaying correctly in the manuscript. Additionally, I prefer not to have five equations presented in a row. The authors might consider including and explaining the variables in Equations 1 and 2 before presenting Equations 3 to 5, which could enhance clarity.

I would also appreciate a citation of at least one previous study that utilizes fixed pricing models based on production costs. Furthermore, a table summarizing all the assumptions in the economic model would be helpful.

The rationale for selecting interest rates from 0% to 30% should be clarified. Are these rates typical for similar projects, or are they hypothetical scenarios? Moreover, the relationship between different rates and the Internal Rate of Return (IRR) should be more explicitly linked to the overall investment strategy.

The conclusions could benefit from clearer organization. Presenting them in a list format is less effective than integrating them into a narrative that outlines the key findings and their implications. A more logical flow would enhance readability.

Overall, these revisions will help strengthen the paper and better convey its contributions to the field.

Reviewer #2: None required - The authors have addressed the questions and concerns about the manuscript, and the revised version is better.

7. PLOS authors have the option to publish the peer review history of their article (what does this mean?). If published, this will include your full peer review and any attached files.

Reviewer #1: **Yes: **Brooke Rogachuk

Reviewer #2: No

---

## [Author Response · Author response to Decision Letter 2]

22 Nov 2024

We would like to start by thanking the respected Academic Editor, respected Reviewer 1, and respected Reviewer 2 for taking the time to review our manuscript and for their variable comments and suggestions. Kindly find below our reply on these comments. Please note that all changes are highlighted in yellow in the revised version of the manuscript.

Reviewer 1

Reviewer 1 comment:

The authors did a great job with the edits! However, there are still a few issues in the paper. Firstly, the figures are not displaying correctly in the manuscript.

Authors reply:

We thank the respected reviewer for reviewing our manuscript and for pointing this out. Kindly note that the figures were provided to the respected journal in tif format, according to the journal instructions. The pdf builder system does not display them in the manuscript in their full resolution, which is not the case in the final article form. Please find below the link to the figures in their full resolution, which will appear in the published version of the article, if accepted.

https://www.dropbox.com/scl/fo/1n8lk26960px0iaq4u4u6/ALzvjqlDnX9AGETxHbRjbcQ?rlkey=wwya649ohowi0ydhdnlvs4fls&st=f8hpgn7l&dl=0

Thank you very much.

Reviewer 1 comment:

Additionally, I prefer not to have five equations presented in a row. The authors might consider including and explaining the variables in Equations 1 and 2 before presenting Equations 3 to 5, which could enhance clarity.

Authors reply:

We thank the respected reviewer for the provided comment, which we agree with and were happy to take into account. Kindly check Pages 8-9. Thank you very much.

Reviewer 1 comment:

I would also appreciate a citation of at least one previous study that utilizes fixed pricing models based on production costs.

Authors reply:

We thank the respected reviewer for pointing this out. Kindly note that the following reference was included as part of our citations in the revised version of the manuscript:

Doi: 10.1016/j.enconman.2024.118110

Kindly check reference 41 in Page 12, and in the reference list section. Thank you very much.

Reviewer 1 comment:

Furthermore, a table summarizing all the assumptions in the economic model would be helpful.

Authors reply:

The authors thank the respected reviewer for their comment, which we were happy to take into consideration in our revised version of the manuscript. Kindly note that a table including a summary of the assumptions used in the economic analysis was included in the revised version of the manuscript. Kindly check Table 3, Page 14.

Thank you very much.

Reviewer 1 comment:

The rationale for selecting interest rates from 0% to 30% should be clarified. Are these rates typical for similar projects, or are they hypothetical scenarios?

Authors reply:

The authors thank the respected reviewer for pointing this out. Kindly note that the selected interest rates are hypothetical. We have included this in our revised version of the manuscript. Please check Page 11 in the modified version of the manuscript.

Thank you very much.

Reviewer 1 comment:

Moreover, the relationship between different rates and the Internal Rate of Return (IRR) should be more explicitly linked to the overall investment strategy.

Authors reply:

The authors thank the respected reviewer for his comment. We have tried to address the respected reviewer comment in Section 3.5, Page 25. The following statement was included in the revised version of our manuscript:

In general, the IRR is inversely proportional to the interest rate, showing its lowest value at i = 30% among all three studied cases for both nanofluid fabrication methods. According to Eq. 9 and Eq.16, increasing the interest rate would result in a higher TAC, and consequently leading to a lower net cash flow. The previous causes the IRR value to decrease, according to Eq. 15

Thank you very much.

Reviewer 1 comment:

The conclusions could benefit from clearer organization. Presenting them in a list format is less effective than integrating them into a narrative that outlines the key findings and their implications. A more logical flow would enhance readability.

Authors reply:

The authors thank the respected reviewer for the comment. Kindly note that the authors have rewriting the conclusion section to compile with the respected reviewer comment, as best as possible. We hope that the newer version is acceptable by the respected reviewer.

Thank you very much.

Reviewer 1 comment:

Overall, these revisions will help strengthen the paper and better convey its contributions to the field.

Authors reply:

The authors thank the respected reviewer for the time and comments provided. We hope that we were able to improve the quality of our manuscript and meet the level of acceptance.

Thank you very much.

Reviewer 2

Reviewer 2 comment:

None required - The authors have addressed the questions and concerns about the manuscript, and the revised version is better.

Authors reply:

We thank the respected reviewer for taking the time reviewing our manuscript and for his respected decision.

Thank you very much.

---

## [Decision Letter · Decision Letter 2]

30 Jan 2025

Techno-economic assessment of effervescent tablet-based nanofluids

PONE-D-24-29775R2

Dear Dr. Ali,

We’re pleased to inform you that your manuscript has been judged scientifically suitable for publication and will be formally accepted for publication once it meets all outstanding technical requirements.

Kind regards,

Jude Okolie, Ph.D.

Academic Editor

PLOS ONE

Additional Editor Comments (optional):

Please ensure the quality of all images are clear and all references are consistent. Provide excel files for all economic calculations and assumptions used in the economic model.

Reviewers' comments:

Reviewer's Responses to Questions

**Comments to the Author**

1. If the authors have adequately addressed your comments raised in a previous round of review and you feel that this manuscript is now acceptable for publication, you may indicate that here to bypass the “Comments to the Author” section, enter your conflict of interest statement in the “Confidential to Editor” section, and submit your "Accept" recommendation.

Reviewer #1: All comments have been addressed

Reviewer #2: All comments have been addressed

2. Is the manuscript technically sound, and do the data support the conclusions?

Reviewer #1: Yes

Reviewer #2: Yes

3. Has the statistical analysis been performed appropriately and rigorously? 

Reviewer #1: Yes

Reviewer #2: Yes

4. Have the authors made all data underlying the findings in their manuscript fully available?

Reviewer #1: Yes

Reviewer #2: Yes

5. Is the manuscript presented in an intelligible fashion and written in standard English?

Reviewer #1: Yes

Reviewer #2: Yes

6. Review Comments to the Author

Reviewer #1: This is great work! The addition of the comments has truly elevated the quality of your work! Best of luck, and keep up the amazing effort—you're doing fantastic!

Reviewer #2: (No Response)

7. PLOS authors have the option to publish the peer review history of their article (what does this mean?). If published, this will include your full peer review and any attached files.

---

## [Editor Report · Acceptance letter]

PONE-D-24-29775R2

PLOS ONE

Dear Dr. Ali,

I'm pleased to inform you that your manuscript has been deemed suitable for publication in PLOS ONE. Congratulations! Your manuscript is now being handed over to our production team.

Kind regards,

on behalf of

Dr. Jude Okolie

Academic Editor

PLOS ONE